# HOW MANY SAMPLES ARE NEEDED TO TRAIN A DEEP NEURAL NETWORK?

**Pegah Golestaneh, Mahsa Taheri & Johannes Lederer**
Department of Mathematics, Computer Science, and Natural Sciences
University of Hamburg
`{pegah.golestaneh,mahsa.taheri,johannes.lederer}@uni-hamburg.de`

## ABSTRACT

Although neural networks have become standard tools in many areas, many important statistical questions remain open. This paper studies the question of how much data are needed to train a ReLU feed-forward neural network. Our theoretical and empirical results suggest that the generalization error of ReLU feed-forward neural networks scales at the rate $1/\sqrt{n}$ in the sample size $n$—rather than the "parametric rate" $1/n$, which could be suggested by traditional statistical theories. Thus, broadly speaking, our results underpin the common belief that neural networks need "many" training samples. Along the way, we also establish new technical insights, such as the first lower bounds of the entropy of ReLU feed-forward networks.

## 1 INTRODUCTION

Neural networks have ubiquitous applications in science and business (Goodfellow et al., 2016; Graves et al., 2013; LeCun et al., 2015; Badrinarayanan et al., 2017). However, our understanding of their statistical properties remains incomplete. For example, a basic yet very important open question is: how many training samples are needed to train a (non-linear) neural network?

Over the past two decades, significant progress has been made deriving upper bounds for the generalization error of neural networks (Anthony & Bartlett, 2009; Arora et al., 2018; Yarotsky, 2017; Nagarajan & Kolter, 2019). Works like Neyshabur et al. (2015); Bartlett et al. (2017); Neyshabur et al. (2017; 2018) first highlight the relationship between the complexity of the network space and the generalization error and then, they bound the corresponding complexity measure. For example, Neyshabur et al. (2015) study the growth of the Rademacher complexity of classes of $\ell_1$-constraint neural networks. Additionally, generalization error bounds for networks with spectral norm constraints are proposed by Neyshabur et al. (2018) (employing a PAC-Bayesian framework) and Bartlett et al. (2017) (employing a margin-based framework) for Lipschitz activation functions. A common limitation of the works mentioned above is that their generalization bounds often exhibit strong dependence—frequently exponential—on either the depth or width of the network. Golowich et al. (2018) address this by providing bounds that avoid direct dependence on network depth for norm-constrained neural networks, with rates that grow only polynomially in depth. More recently, Taheri et al. (2021; 2022); Mohades & Lederer (2023); Lederer (2023) have proposed generalization bounds for regularized neural networks that show logarithmic growth in the total number of parameters, with the potential to decrease with additional layers, in contrast to exponential growth. For example, Taheri et al. (2021) provides an upper bound for the generalization error of $\ell_1$-regularized feed-forward neural networks for general activation functions growing by $(L/2)^{1/2-L}\sqrt{\log(P)}(\log(n)/\sqrt{n})$, where $P$ corresponds to the total number of parameters in the network, $L$ is the number of hidden layers, and $n$ is the number of training samples.

While works mentioned above target to upper bound the generalization error of neural networks, another line of research study lower bounds for the mini-max risk of neural networks in a parametric setting. The central question here is if improving the rate $1/\sqrt{n}$ is possible for neural networks in general or not. Klusowski & Barron (2017) provide a mini-max lower bound for shallow neural networks with Sinusoidal activation, assuming input vectors are uniformly distributed across $[-1, 1]^d$ that decays as $1/\sqrt{n}$. Du et al. (2018) provide a mini-max lower bound for a CNN with linear activation, where the input domain is normally distributed with a mean of 0 and an identity covariance

matrix. They also determined the sample complexity for a convolutional neural network with linear activation functions and asserted that tackling the challenges posed by non-linear activation functions, even without convolution structure, remains a difficult task.

To make it short, to the best of our knowledge, a **mini-max lower bound** for **deep** feed-forward neural networks with **non-linear activation functions** that **match upper bounds** have not been established. In this paper, we use a technique from information theory known as Fano's inequality to establish such a bound. Our bound scales as $\sqrt{\log(d)/n}$ (with $n$ as the number of training samples and $d$ as the input dimension) that matches the above-mentioned upper bound rate in Taheri et al. (2021). We also confirm this rate in several applications for both regression and classification tasks. Thus, we demonstrate that the minimum number of samples needed to train a neural network to achieve a prediction error $\epsilon^2$ essentially scales as $1/\epsilon^4$—rather than $1/\epsilon^2$. One of the main challenges in our proofs is establishing a sharp lower bound for the packing number of deep ReLU neural networks, which, to the best of our knowledge, is being extracted for the first time in our paper (see Lemma 3.2).

**Our key technical contributions are:**

1. We establish a mini-max lower bound for the risk of ReLU feed-forward networks that depends on the input dimensions only through a logarithmic factor and the level of sparsity of the parameter space. The bound decreases as $1/\sqrt{n}$ with the number of training samples $n$, which matches recent upper bounds. It is also important to note that the rate of $1/2$ in the exponent remains the same across various input dimensions and network configurations (Theorem 2.2).

2. We support this theoretical bound empirically, also beyond feed-forward settings to include CNNs (Section 5).

3. We establish a lower bound for the packing number of deep ReLU networks' spaces (Lemma 3.2).

**The main broader implication is as follows:**

- Most practitioners are convinced that deep learning requires "more" data than classical methods. It is well-known that the accuracy of classical methods, like the least-squares estimator in linear regression, scale like $1/n$ in the sample size (Wainwright, 2019; Guo1 et al., 2024). Our mathematical proof along with the numerical illustrations showing that this is not the case for (non-linear) deep learning, are arguably the first confirmation and underpinning of the mentioned practitioners' belief.

More generally, the paper provides a rigorous mathematical perspective on an important aspect of deep learning to complement the much more abundant "engineering perspective" in the field. We will come back to the relevance of this perspective in the conclusion section (Section 6).

**Other related works:** In addition to the aforementioned works, a growing body of research has emerged focused on employing deep ReLU networks to approximate non-parametric regression models (Suzuki, 2019; Parhi & Nowak, 2022; Raskutti et al., 2009; Schmidt-Hieber & Bos, 2022; Raskutti et al., 2012; Schmidt-Hieber, 2020; Zhang & Wang, 2023; Tsuji, 2021). These models are characterized by sparse additive structures and specific smoothness properties, such as Hölder, Besov, or Sobolev functions. These studies investigate the approximation capabilities of deep ReLU networks for non-parametric regression functions by establishing mini-max optimal rates. It is crucial to consider that their mini-max lower bounds for the function classes degrade either with the depth of the network or with the parameters of smoothness.

In this paper, our perspective differs slightly from that of others, such as Schmidt-Hieber (2020), as we view deep ReLU networks as fundamental functions of interest in their own right, rather than as approximation methods for other function classes. We explore the statistical properties of deep ReLU networks by establishing a mini-max lower bound for these networks. In contrast, they emphasize the distinction between function classes and approximation methods and aim to confirm the superiority of deep learning to other representative methods like wavelet transforms, kernel methods, and spline methods in non-parametric settings. Schmidt-Hieber (2020) shows that sparse deep neural networks with ReLU activation function and a well-designed architecture achieve the

mini-max rates of convergence (up to log n-factors) in multivariate non-parametric regression, under a broad composition assumption on the regression function. They focus on the regression setup and demonstrate that deep ReLU networks can achieve faster rates under a hierarchical composition assumption on the regression function. This assumption encompasses (generalized) additive models as well as the composition models considered in (Juditsky et al., 2009; Kohler & Krzyżak, 2017). Furthermore, they demonstrate that, given the composition assumption, wavelet estimators are only able to obtain sub-optimal rates. Theoretical analysis in Suzuki (2019) reveals that deep learning with ReLU activation in Besov spaces exhibits superior adaptivity, demonstrating mini-max optimal rates and outperforming non-adaptive estimators such as kernel ridge regression. Furthermore, the study highlights deep learning's ability to mitigate the curse of dimensionality in mixed smooth Besov spaces, emphasizing its high adaptivity and effectiveness as a feature extractor in capturing the spatial inhomogeneity of target functions. Imaizumi & Fukumizu (2019) provides a comprehensive review of prior research related to function estimation using deep neural networks.

Thus, to the best of our knowledge, our work is the first to illustrate, both theoretically and practically, that in the absence of additional assumptions, the rate of $1/\sqrt{n}$ is the "correct rate" in deep learning. It is important to note that some works, such as Schmidt-Hieber (2020), obtain a rate of $1/n$, but only under very strict assumptions. For instance, to achieve a rate of $1/n$ in their results, it is necessary to assume that the parameters are bounded by one and the network functions are bounded.

**Organisation:** Section 2 provides the problem formulation and establishes a lower bound on the mini-max risk for ReLU feed-forward neural networks (Theorem 2.2). Section 3 provides some technical results that form our main result's foundation including, a lower bound for the packing number of a ReLU network's space (Lemma 3.2). We provide the proof of our main theorem in Section 4. In Section 5, we shift our focus to the empirical findings to support our theories. We conclude our paper in Section 6. More technical results, empirical details, and detailed proofs are deferred to the Appendix.

## 2 PROBLEM FORMULATION AND MAIN RESULT

This section provides an outline of the core elements of our study. We introduce the problem setting before presenting our main result. To start, we consider the following regression model

$$y_i = f^*(\boldsymbol{x}_i) + u_i \qquad \text{for } i \in \{1, \dots, n\}, \tag{1}$$

for an unknown neural network $f^* : \mathbb{R}^d \to \mathbb{R}$ and independent and identically distributed noise $u_i \sim \mathcal{N}(0, \sigma^2)$ with $\sigma \in (0, \infty)$. We observe $n$ independent and identically distributed data samples $(\boldsymbol{x}_1, y_1), (\boldsymbol{x}_2, y_2), \dots, (\boldsymbol{x}_n, y_n) \in \mathbb{R}^d \times \mathbb{R}$ drawn independently from a joint distribution $\mathbb{P}_{\boldsymbol{x}, y}$ with a fixed marginal distribution $\mathbb{P}_{\boldsymbol{x}} = \mathcal{N}(\boldsymbol{0}, \boldsymbol{I}_d)$. It is assumed that $u_i$ and $\boldsymbol{x}_i$ are independent and that the networks are of the form

$$
\begin{aligned}
f_{\boldsymbol{\Theta}} &: \mathbb{R}^d \to \mathbb{R} \\
\boldsymbol{x} &\mapsto f_{\boldsymbol{\Theta}}(\boldsymbol{x}) := W^L \phi^L\big(\dots W^1 \phi^1(W^0 \boldsymbol{x})\big)
\end{aligned}
\tag{2}
$$

indexed by $\boldsymbol{\Theta} = (W^L, \dots, W^0)$ summarizing the weight matrices $W^l \in \mathbb{R}^{h_{l+1} \times h_l}$ for $l \in \{0, 1, \dots, L\}$. The number of hidden layers (the depth of the network) denotes as $L \in \{1, 2, \dots\}$, and $h_l$ denotes the number of nodes in the $l$-th layer (the width of the $l$-th layer), where $h_0 = d$ and $h_{L+1} = 1$. The function $\phi^l : \mathbb{R}^{h_l} \to \mathbb{R}^{h_l}$ is the ReLU activation function of the $l$-th layer, and for a vector $\boldsymbol{z} = [z_1, \dots, z_{h_l}] \in \mathbb{R}^{h_l}$ is defined as

$$\phi^l(\boldsymbol{z}) := \big[\max\{0, z_1\}, \max\{0, z_2\}, \dots, \max\{0, z_{h_l}\}\big].$$

We then consider a ($\ell_1$-type) sparse parameter space $\mathcal{B}_L$ constraints on the parameters of the network. We use $\ell_1$-type constraints as opposed to $\ell_0$-type constraints, primarily because $\ell_0$-type constraints tend to make the problem hard to optimize and "combinatorial", particularly in high-dimensional settings (Lederer, 2022, Chapter 2). Furthermore, although the $\ell_1$-type constraint is non-smooth, it generally causes few computational issues (see Friedman et al. (2010)). Also, it has recently been successfully applied to encourage sparsity in neural networks (Lemhadri et al., 2021). Then, we define a function class

$$\mathcal{F}_{\mathcal{B}_L} := \left\{ f_{\boldsymbol{\Theta}} \mid \boldsymbol{\Theta} = (W^L, \dots, W^0) \in \mathcal{B}_L \right\},$$

where $\mathcal{B}_{\mathrm{L}}$, denotes a sparse parameter space for ReLU feed-forward networks and defined as

$$\mathcal{B}_{\mathrm{L}} := \left\{ (W^L, \ldots, W^0) \mid \sum_{l=1}^{L} \|\!|W^l|\!\|_1 \leq v_{\mathrm{s}}, \ \|\!|W^0_{j,\cdot}|\!\|_1 \leq v_0 \ \text{for all} \ \ j \in \{1, \ldots, h_1\} \right\},$$

for $W^l \in \mathbb{R}^{h_{l+1} \times h_l}$ and $l \in \{0, 1, \ldots, L\}$ and we define $\|\!|W^l|\!\|_1 := \sum_{k=1}^{h_{l+1}} \sum_{j=1}^{h_l} |W^l_{kj}|$.

**Assumption 2.1** ($v_0 = 1$) *For simplicity in the proof of our technical results (Lemma 3.2), we assume $v_0 = 1$.*

This assumption is useful for constructing a subclass of space $\mathcal{F}_{\mathcal{B}_{\mathrm{L}}}$ in the proof of Lemma 3.2 to establish a lower bound for the packing number of a deep ReLU network's space. This assumption basically determines the structure of the weight between the input layer and the first hidden layer of a neural network. It also guarantees that the number of input dimensions $d$, matches the width of the constructed subclass of $\mathcal{F}_{\mathcal{B}_{\mathrm{L}}}$.

The mini-max risk for the function class $\mathcal{F}_{\mathcal{B}_{\mathrm{L}}}$ can be defined as (Wainwright, 2019, Chapter 15)

$$\mathcal{R}_{(n,d)}(\mathcal{F}_{\mathcal{B}_{\mathrm{L}}}; \Phi \circ \rho) := \inf_{\widehat{f}} \sup_{f^* \in \mathcal{F}_{\mathcal{B}_{\mathrm{L}}}} \mathbb{E}_{(\boldsymbol{x}_i, y_i)_{i=1}^n} \left[ \Phi\big(\rho(\widehat{f}, f^*)\big) \right], \tag{3}$$

where $\rho : \mathcal{F}_{\mathcal{B}_{\mathrm{L}}} \times \mathcal{F}_{\mathcal{B}_{\mathrm{L}}} \to [0, \infty)$ is a semi metric[1] and $\Phi : [0, \infty) \to [0, \infty)$ is an increasing function. The expectation is taken with respect to the training data $(\boldsymbol{x}_i, y_i)_{i=1}^n$ and the infimum runs over all possible estimators $\widehat{f}$ (measurable functions) of $f^*$ on the training data $(\boldsymbol{x}_i, y_i)_{i=1}^n$. Hence, $\widehat{f}(\boldsymbol{x}) \equiv \widehat{f}(\boldsymbol{x}, \{(\boldsymbol{x}_i, y_i)\}_{i=1}^n)$, where $\boldsymbol{x}$ is a new data point with the same distribution $\mathbb{P}_{\boldsymbol{x}}$. We use the notation $\mathcal{R}_{(n,d)}(\mathcal{F}_{\mathcal{B}_{\mathrm{L}}}; \Phi \circ \rho)$ to emphasize that the mini-max risk depends on the number of training samples $n$, the input dimension $d$ and the space $\mathcal{F}_{\mathcal{B}_{\mathrm{L}}}$.

In this paper, our focus is on the standard setting where $\rho$ represents the $L_2(\mathbb{P}_{\boldsymbol{x}})$-norm, and $\Phi(t) = t^2$. Therefore, $\Phi(\rho(\widehat{f}, f^*))$ is the squared $L_2(\mathbb{P}_{\boldsymbol{x}})$-norm, that is our mini-max risk

$$\inf_{\widehat{f}} \sup_{f^* \in \mathcal{F}_{\mathcal{B}_{\mathrm{L}}}} \mathbb{E}_{(\boldsymbol{x}_i, y_i)_{i=1}^n} \left[ \|\widehat{f} - f^*\|_{L_2}^2 \right].$$

We assume that the distribution $\mathbb{P}_{\boldsymbol{x}}$ has a density $h(\boldsymbol{x})$ with respect to the Lebesgue measure $d\boldsymbol{x}$ which, implies that

$$\|\widehat{f} - f^*\|_{L_2} := \left( \int_{\boldsymbol{x} \in \mathcal{X}} \big(\widehat{f}(\boldsymbol{x}) - f^*(\boldsymbol{x})\big)^2 h(\boldsymbol{x}) d\boldsymbol{x} \right)^{1/2}.$$

We now present our mini-max risk lower bound for deep ReLU neural networks. Considering the regression model defined in Equation (1), where $f^* \in \mathcal{F}_{\mathcal{B}_{\mathrm{L}}}$ (a ReLU neural network with $L$ hidden layers and $\ell_1$-bounded weights), then we have:

**Theorem 2.2 (Mini-max risk lower bound for ReLU feed-forward networks)** *Using the $L_2(\mathbb{P}_{\boldsymbol{x}})$-norm as our underlying semi metric $\rho$, and $\boldsymbol{x}_1, \ldots, \boldsymbol{x}_n \sim \mathcal{N}(\boldsymbol{0}, \boldsymbol{I}_d)$, then for $d$ large enough and any increasing function $\Phi : [0, \infty) \to [0, \infty)$, it holds that*

$$\mathcal{R}_{(n,d)}(\mathcal{F}_{\mathcal{B}_{\mathrm{L}}}; \Phi \circ \rho) \geq \frac{1}{2} \Phi \left[ c\sqrt{V_{\mathcal{F}}} \left(\frac{\log(d)}{n}\right)^{1/4} \right], \tag{4}$$

*with $c := \sqrt{\tau \sigma / 160}$, where $\tau \in (0, \infty)$ is a numerical constant, and $V_{\mathcal{F}} := (v_{\mathrm{s}}/L)^L$. For $\Phi(\cdot) = (\cdot)^2$, we specifically obtain*

$$\inf_{\widehat{f}} \sup_{f^* \in \mathcal{F}_{\mathcal{B}_{\mathrm{L}}}} \mathbb{E}_{(\boldsymbol{x}_i, y_i)_{i=1}^n} \left[ \|\widehat{f} - f^*\|_{L_2}^2 \right] \geq \frac{c^2}{2} (V_{\mathcal{F}}) \sqrt{\frac{\log(d)}{n}}. \tag{5}$$

---

[1]A semi metric satisfies all properties of a metric, except that there may exist pairs $f \neq f'$ for which $\rho(f, f') = 0$.

For the technical reasons, we assume that the input dimension is large enough, let say $d \in [10, \infty)$. Our mini-max lower bound above reveals essentially a $1/\sqrt{n}$-decrease in the sample size, and a $\log(d)$-increase in the input dimension, that perfectly aligns with the upper bounds on the generalization error of deep neural networks with $\ell_1$-type regularization (compare to Taheri et al. (2021)). Additionally, the factor $V_{\mathcal{F}}$ in our rates can be interpreted as a product over the $\ell_1$-norm bounds of different layers in $\mathcal{F}_{\mathcal{B}_L}$. This interpretation aligns perfectly with the established upper and lower bounds on the generalization error of regularized neural networks found in the literature (Taheri et al., 2021; Klusowski & Barron, 2017). The presence of $V_{\mathcal{F}}$ in our rate supports the sharpness of our result (compare with Taheri et al. (2021)).

Theorem 2.2 demonstrates that for all possible $\widehat{f}$, risk (also called "generalization error") scales at least as $V_{\mathcal{F}} \sqrt{\log(d)/n}$. Then, by considering an upper bound for the risk

$$\mathbb{E}_{(\boldsymbol{x}_i, y_i)_{i=1}^n} \left[ \|\widehat{f} - f^*\|_{L_2}^2 \right] \leq \epsilon^2 \,,$$

the result of Theorem 2.2 can be reformulated. It implies that one requires a minimum of

$$n \geq \left( \frac{c}{\epsilon} \right)^4 \frac{(V_{\mathcal{F}})^2 \log(d)}{4} \,, \tag{6}$$

samples to achieve an error of at most $\epsilon^2$ for ReLU feed-forward neural networks. Based on this formula, while we can't determine the exact required number of samples in advance, we can still make some useful observations.

1. We need "many" samples based on the rate $1/\sqrt{n}$.

2. The noisier the data, the more training samples are needed to achieve the desired error level.

This also stands in strong contrast to classical methods (consider, for instance, linear regression with the least squares estimator as the quintessential example of a traditional pipeline), which usually require of the order $1/\epsilon^2$ samples to reach the same accuracy. However, of course, this does not mean that one should not apply deep learning! Deep learning is extremely flexible in approximating functions, which makes it the preferred method in many modern applications.

## 3 TECHNICAL RESULTS

Here, we provide a technical result essential in proving our main theorem 2.2. Since our proof approach is based on a classical result from information theory known as Fano's inequality (Lemma A.1), we need to find a lower bound for the packing number of our network's space (Lemma 3.2).

The following notation will be used throughout the paper. For a vector $\boldsymbol{v} \in \mathbb{R}^d$, $\ell_0$-norm is defined by $\|\boldsymbol{v}\|_0 := \#\{i \in \{1, \ldots, d\} : v_i \neq 0\}$, $\ell_1$-norm is defined by $\|\boldsymbol{v}\|_1 := \sum_{i=1}^d |v_i|$ and the Euclidean norm is defined by $\|\boldsymbol{v}\|_2 := \sqrt{\sum_{i=1}^d (v_i)^2}$. We recall $\|W^l\|_1 := \sum_{k=1}^{h_{l+1}} \sum_{j=1}^{h_l} |W_{kj}^l|$ and $\|W^l\|_\infty := \max_{1 \leq k \leq h_{l+1}} \sum_{j=1}^{h_l} |W_{kj}^l|$ for a matrix $W^l \in \mathbb{R}^{h_{l+1} \times h_l}$. The cardinality of the $2\delta$-packing of the corresponding network's space $\mathcal{F}_{\mathcal{B}_L}$ for $\delta \in (0, \infty)$ and with respect to $L_2(\mathbb{P}_{\boldsymbol{x}})$-norm is defined as $\mathcal{M} := \mathcal{M}(2\delta, \mathcal{F}_{\mathcal{B}_L}, \|\cdot\|_{L_2})$. We use the notation $D_{\mathrm{KL}}(\mathbb{P} \parallel \mathbb{Q})$ to denote the Kullback-Leibler (KL) divergence between two probability distributions $\mathbb{P}$ and $\mathbb{Q}$. We define $[\mathcal{M}] := \{1, \ldots, \mathcal{M}\}$ as the index set. And we define the notation $X^n := (\boldsymbol{x}_1, \ldots, \boldsymbol{x}_n)^\top$ and $Y^n := (y_1, \ldots, y_n)^\top$.

We now present the definition of the packing number, as given in (Vaart & Wellner, 1996).

**Definition 3.1 (Packing number)** *Consider a metric space consisting of a set $\mathcal{F}_{\mathcal{B}_L}$ and a semi metric $\rho$ as defined in Section 2 then, an $2\delta$-packing of $\mathcal{F}_{\mathcal{B}_L}$ in the semi metric $\rho$ is a collection $\{f_{\Theta^1}, \ldots, f_{\Theta^{\mathcal{M}}}\} \subseteq \mathcal{F}_{\mathcal{B}_L}$ such that $\rho(f_{\Theta^j}, f_{\Theta^k}) \geq 2\delta$ for all $j, k \in [\mathcal{M}]$ and $j \neq k$. The $2\delta$-packing number $\mathcal{M}(2\delta, \mathcal{F}_{\mathcal{B}_L}, \rho)$, is the cardinality of the largest $2\delta$-packing.*

Our next lemma provides a lower bound for the packing number of a ReLU network's space.

**Lemma 3.2 (Lower bounding the packing numbers of ReLU feed-forward networks' spaces)** *For a sparse collection of deep ReLU feed-forward networks' spaces $\mathcal{F}_{\mathcal{B}_L}$, there exist $\delta \in (0, \infty)$*

*such that*

$$\log \mathcal{M}\big(2\delta, \mathcal{F}_{\mathcal{B}_{\mathrm{L}}}, \|\cdot\|_{L_2}\big) \geq \Big(\frac{\tau V_{\mathcal{F}}}{20\delta}\Big)^2 \log(d)\,,$$

*where $\tau \in (0,\infty)$ is a numerical constant and $V_{\mathcal{F}} = (v_{\mathrm{s}}/L)^L$.*

This lemma provides valuable insights into the capacity and potential complexity of ReLU feed-forward networks. The key components of this bound are twofold: 1. the favorite $\log(d)$ factor (and not the huge network size) and 2. the factor $V_{\mathcal{F}}$, that both are enhanced employing the $\ell_1$-norm control over the parameters of the network. In particular, the logarithmic dependence on the input dimension $d$ and the factor $V_{\mathcal{F}}$ offer a perspective on how complexity growth relates to the network size and the sparsity level.

## 4 PROOF OF THEOREM 2.2

Here, we provide the proof of our main theorem:

**Proof**  The proof is based on employing a variants of Fano's method well-known as "local packing" or "rescaling" (Wainwright, 2019; Yang & Barron, 1999) and our Lemma 3.2.

Let's start the proof writing the mutual information in terms of KL divergence (see also Section A): assuming a $2\delta$-packing is available with centers $j \in [\mathcal{M}]$; then, using Wainwright (2019, Equation 15.30) we can obtain

$$I(J; Y^n|X^n) = \frac{1}{\mathcal{M}} \sum_{j=1}^{\mathcal{M}} D_{\mathrm{KL}}(\mathbb{P}^n_{f_{\Theta^j}} \parallel \bar{\mathbb{Q}}) \leq \max_{j,k \in [\mathcal{M}], j \neq k} D_{\mathrm{KL}}(\mathbb{P}^n_{f_{\Theta^j}} \parallel \mathbb{P}^n_{f_{\Theta^k}})\,,$$

where $\bar{\mathbb{Q}} := (1/\mathcal{M})\sum_{j=1}^{\mathcal{M}} \mathbb{P}^n_{f_{\Theta^j}}$ is the mixture distribution, $J$ is uniformly distributed over the index set $[\mathcal{M}]$, and $\mathbb{P}^n_{f_{\Theta^j}}$ is the $n$-product distribution (see Section A). The naive idea of "local packing" is considering a $2\delta$-packing within the space $\mathcal{F}_{\mathcal{B}_{\mathrm{L}}}$ (in the semi metric $\rho$) such that $\max_{j,k \in [\mathcal{M}], j \neq k} D_{\mathrm{KL}}(\mathbb{P}^n_{f_{\Theta^j}} \parallel \mathbb{P}^n_{f_{\Theta^k}}) \leq n(2\kappa\delta)^2$ for a quantity $\kappa \in [1,\infty)$. That implies the KL divergence should be bounded by a multiple of $\delta$ for all centers in the packing set. In fact, instead of the whole space, we consider just a local area of that (specified as a ball with radius $2\kappa\delta$). Then, employing our earlier display implies an upper bound for the mutual information.

Following above technique, we focus on a local area of the network's space and call it $\mathcal{F}_S$, consider a $2\delta$-packing set of that where for two distinct networks $f_{\Theta^j}, f_{\Theta^k} \in \mathcal{F}_S$ with $j, k \in [\mathcal{M}']$ we have $\rho(f_{\Theta^j}(\boldsymbol{x}), f_{\Theta^k}(\boldsymbol{x})) \geq 2\delta$. Also, we suppose that $\rho(f_{\Theta^j}(\boldsymbol{x}), f_{\Theta^k}(\boldsymbol{x})) \leq 2\kappa\delta$ for a constant $\kappa \in [1,\infty)$ for $f_{\Theta^j}, f_{\Theta^k} \in \mathcal{F}_S$ with $j, k \in [\mathcal{M}']$ (note that we can always find such an area just by rescaling across our networks output; exact value of $\delta$ be specified later). Then, employing Lemma A.3, we obtain

$$D_{\mathrm{KL}}(\mathbb{P}^n_{f_{\Theta^j}} \parallel \mathbb{P}^n_{f_{\Theta^k}}) \leq \frac{n}{2\sigma^2}\rho\big(f_{\Theta^j}(\boldsymbol{x}), f_{\Theta^k}(\boldsymbol{x})\big)^2 \leq \frac{n}{2\sigma^2}(2\kappa\delta)^2\,,$$

for all $j \neq k \in [\mathcal{M}']$, where the last inequality is reached using our assumption on $\mathcal{F}_S$, that is a ball with radius $2\kappa\delta$. Collecting results above, we can obtain $I(J; Y^n|X^n) \leq n(2\kappa\delta)^2/2\sigma^2$ (see also Lemma A.5).

In the second step and based on (local) Fano's inequality, we need to specify $2\delta$-packing set with the largest cardinality to verify $(I(J; Y^n|X^n) + \log 2)/\log \mathcal{M} \leq 1/2$. So we assign the value of $\delta$ to ensure

$$\log \mathcal{M}\big(2\delta, \mathcal{F}_S, \|\cdot\|_{L_2}\big) \geq \Big(\frac{n(2\kappa\delta)^2}{\sigma^2} + 2\log 2\Big) \geq \Big(\frac{4n(\kappa\delta)^2}{\sigma^2}\Big).$$

Next step is lower bounding $\log \mathcal{M}(2\delta, \mathcal{F}_S, \|\cdot\|_{L_2})$. Let recall that $\mathcal{F}_S$ is a local area of $\mathcal{F}_{\mathcal{B}_{\mathrm{L}}}$ ($\mathcal{F}_S \subset \mathcal{F}_{\mathcal{B}_{\mathrm{L}}}$). Employing Yang & Barron (1999, Lemma 3), $\log \mathcal{M}(2\delta, \mathcal{F}_S, \|\cdot\|_{L_2})$ that is a local-entropy can be lower-bounded on a high level by a fraction of $\log \mathcal{M}(2\delta, \mathcal{F}_{\mathcal{B}_{\mathrm{L}}}, \|\cdot\|_{L_2})$ that is the global entropy. Accordingly, we employ our lower bound on the packing number in Lemma 3.2 as a lower bounding (a fraction of) $\log \mathcal{M}(2\delta, \mathcal{F}_{\mathcal{B}_{\mathrm{L}}}, \|\cdot\|_{L_2})$ that implies

$$\frac{4}{\sigma^2} n(\kappa\delta)^2 = \Big(\frac{\tau V_{\mathcal{F}}}{20\delta}\Big)^2 \log(d)\,,$$

and gives

$$\delta^2 = \frac{\tau \sigma V_{\mathcal{F}}}{40\kappa} \sqrt{\frac{\log(d)}{n}} \, .$$

Let note that our specified value for $\delta$ above is not exact, because for simplicity we just keep the impact of main factors like $n, d$ and $\sigma$, while skipping some constants connecting the local and global entropy. We can 1. substitute the obtained value of $\delta$ into "local packing" version of Fano's inequality (Wainwright, 2019, Section 15.3.3), 2. perform some rewriting and 3. plug in the value of $(\kappa = 4)$ (see Appendix Section A) to yield

$$\mathcal{R}_{(n,d)}(\mathcal{F}_{\mathcal{B}_{\mathrm{L}}}; \Phi \circ \rho) \geq \frac{1}{2} \Phi \left[ \left( \frac{\tau \sigma V_{\mathcal{F}}}{40\kappa} \sqrt{\frac{\log(d)}{n}} \right)^{1/2} \right]$$

$$= \frac{1}{2} \Phi \left[ \left( \frac{\tau \sigma V_{\mathcal{F}}}{40\kappa} \right)^{1/2} \left( \frac{\log(d)}{n} \right)^{1/4} \right]$$

$$= \frac{1}{2} \Phi \left[ \left( \frac{\tau \sigma V_{\mathcal{F}}}{160} \right)^{1/2} \left( \frac{\log(d)}{n} \right)^{1/4} \right] .$$

We can plug the value of $c$— in the view of Theorem 2.2— into this inequality and get

$$\mathcal{R}_{(n,d)}(\mathcal{F}_{\mathcal{B}_{\mathrm{L}}}; \Phi \circ \rho) \geq \frac{1}{2} \Phi \left[ c \sqrt{V_{\mathcal{F}}} \left( \frac{\log(d)}{n} \right)^{1/4} \right] ,$$

which proves our first claim. For the second claim, we simply use $\Phi(\cdot) = (\cdot)^2$ to obtain

$$\mathcal{R}_{(n,d)}(\mathcal{F}_{\mathcal{B}_{\mathrm{L}}}; \Phi \circ \rho) \geq \frac{c^2}{2} (V_{\mathcal{F}}) \sqrt{\frac{\log(d)}{n}} \, .$$

Based on our mini-max risk setting (Section 2), the above expression can be presented as follows:

$$\inf_{\widehat{f}} \sup_{f^* \in \mathcal{F}_{\mathcal{B}_{\mathrm{L}}}} \mathbb{E}_{(\boldsymbol{x}_i, y_i)_{i=1}^n} \left[ \| \widehat{f} - f^* \|_{L_2}^2 \right] \geq \frac{c^2}{2} (V_{\mathcal{F}}) \sqrt{\frac{\log(d)}{n}} \, ,$$

as desired. ∎

## 5 EMPIRICAL STUDIES

This section supports our theoretical findings with simulations on benchmark datasets. While our theoretical results were established based on feed-forward ReLU neural networks, in this empirical study, we extend our investigation to include both ReLU feed-forward networks and ReLU CNNs. The focus is on understanding whether the generalization error of ReLU neural networks scales more significantly with a $1/n$-rate or a $1/\sqrt{n}$-rate. We use prediction error of our test samples as an estimate of the "generalization error" (as defined in Section 2) for the trained ReLU networks.

For our experiments, we consider both classification and regression tasks. The datasets used include `MNIST`, `Fashion-MNIST` and `CIFAR10` for classification, and the California Housing Prices (CHP) dataset for regression analysis. We use `Cross-entropy` (CE) and `Mean-squared` (MS) error as loss functions for classification and regression datasets, respectively. The implementation of these neural networks was carried out using the `TensorFlow` library (see Appendix C for further details). It is also important to note that for none of these real-world datasets do we consider any assumptions such as Gaussian distributions or other assumptions. We conduct our experiments in two steps: in the first step, we train the network and compute the error for the test samples. In the second step, we determine the appropriate curve (either $1/\sqrt{n}$ or $1/n$ scales) that best fits the test error values. To address the impact of various factors like network depth ($L$), the number of parameters ($P$) and the width of the hidden layers, we consider two curves $(c_1 + \alpha/\sqrt{n})$ and $(c_2 + \beta/n)$ with $\alpha, \beta, c_1, c_2 \in (0, \infty)$ (note that we consider the constant terms $c_1, c_2$ in the context of approximation error). Optimizing these parameters is achieved through the `Sequential Least Squares Quadratic Programming` (SLSQP) method (Kraft, 1988) and the `minimize` function from

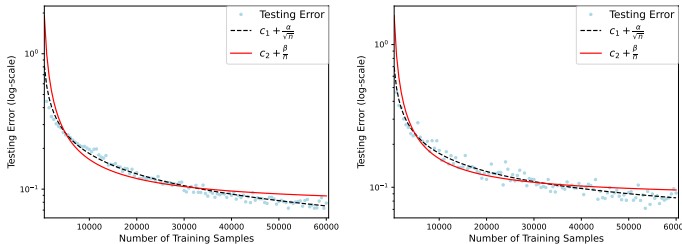

Figure 1: Comparative analysis of the strength of two curves $(c_1 + \alpha/\sqrt{n})$ and $(c_2 + \beta/n)$ to model the generalization error in different architectures for the `MNIST` dataset: (a) Shallow ReLU feed-forward network, width of 128 and (b) Four-hidden-layer network, uniform width of 128 (left to right).

`scipy.optimize` is employed for `SLSQP` implementation. The objective function calculates the sum of squared differences between the generalization error of a neural network and two separate curves. The optimization process aims to find the values for the coefficients that minimize this sum of squared differences. We investigate the strength of these two curves in fitting the generalization error behavior both visually, through figures (Figure 1 to Figure 5), and numerically, using metrics such as R-squared and MSE values (Table 1 to Table 4). We begin this investigation with the `MNIST` dataset.

**MNIST** The `MNIST` dataset consists of $60\,000$ training images and $10\,000$ testing images, each with dimensions of $28 \times 28$ pixels. We perform experiments by incrementally increasing the number of training samples in steps of 500, evaluating the performance across various neural network architectures, including both shallow and deep ReLU feed-forward networks. We consider two scenarios in our analysis. In the first scenario, we consider the entire dataset and gradually increase the number of training samples in intervals of 500. In the second scenario, we focus on larger training samples—we begin with $20\,000$ training samples— and similarly increase the sample size in intervals of 500. Figure 1(a) and Figure 2(a) illustrate the result of our analysis for a shallow neural network with a width of 128, while Figure 1(b) and Figure 2(b) illustrate the result for a four-hidden layer ReLU feed-forward neural network with a uniform width of 128.

Table 1: Numerical metrics to compare the strength of two curves $(c_1 + \alpha/\sqrt{n})$ and $(c_2 + \beta/n)$ to model the generalization error in different architectures for the `MNIST` dataset.

|  | MSE | | R-squared | |
|---|---|---|---|---|
| Architectures | $(c_2 + \beta/n)$ | $(c_1 + \alpha/\sqrt{n})$ | $(c_2 + \beta/n)$ | $(c_1 + \alpha/\sqrt{n})$ |
| Shallow ReLU Network | $1.7 \cdot 10^{-2}$ | $5.2 \cdot 10^{-4}$ | $-1.4 \cdot 10^{0}$ | $9.3 \cdot 10^{-1}$ |
| Four-hidden-layer ReLU Network | $1.1 \cdot 10^{-2}$ | $2.4 \cdot 10^{-4}$ | $-7.8 \cdot 10^{-1}$ | $9.6 \cdot 10^{-1}$ |

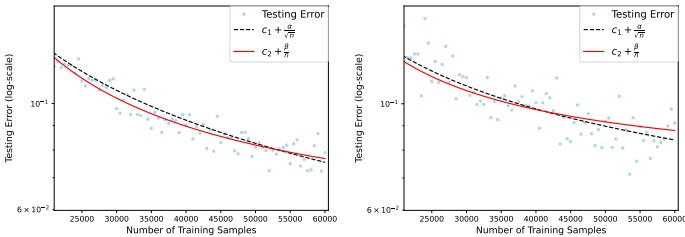

Figure 2: Comparative analysis of the strength of two curves $(c_2 + \beta/n)$ and $(c_1 + \alpha/\sqrt{n})$ to model the generalization error in different architectures for the `MNISTS` dataset for larger training samples: (a) Shallow ReLU feed-forward network, width of 128 and (b) Four-hidden-layer network, uniform width of 128 (left to right).

**Fashion-MNIST** The `Fashion-MNIST` dataset contains $60\,000$ training images and $10\,000$ testing images, both with dimensions of $28 \times 28$ pixels. We run the experiments in intervals of 500

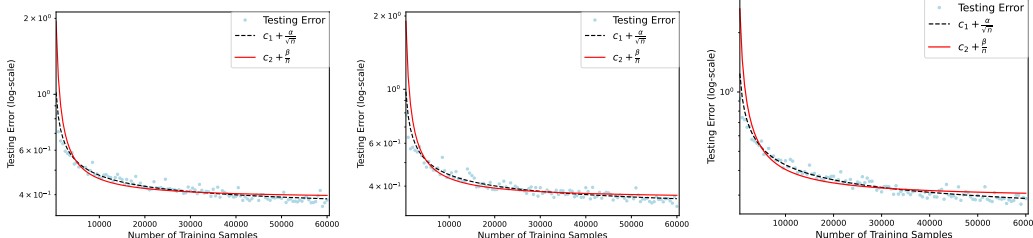

Figure 3: Comparative analysis of the strength of two curves $(c_1 + \alpha/\sqrt{n})$ and $(c_2 + \beta/n)$ to model the generalization error in different architectures for the `Fashion-MNIST` dataset: (a) Shallow ReLU feed-forward network, width of 100, (b) Three-hidden-layer network, uniform width of 100, and (c) CNN ReLU network (left to right).

samples (training samples) and examined with three different architectures. Figure 3(a) illustrates the result of our analysis for a shallow neural network with a width of 100, while Figure 3(b) illustrates the result for a three-hidden layer ReLU feed-forward neural network with a uniform width of 100. Additionally, Figure 3(c) illustrates the result for a CNN ReLU.

Table 2: Numerical metrics to compare the strength of two curves $(c_1 + \alpha/\sqrt{n})$ and $(c_2 + \beta/n)$ to model the generalization error in different architectures for the `Fashion-MNIST` dataset.

| | MSE | | R-squared | |
|---|---|---|---|---|
| Architectures | $(c_2 + \beta/n)$ | $(c_1 + \alpha/\sqrt{n})$ | $(c_2 + \beta/n)$ | $(c_1 + \alpha/\sqrt{n})$ |
| Shallow ReLU Network | $1.3 \cdot 10^{-2}$ | $6.1 \cdot 10^{-4}$ | $-1.6 \cdot 10^0$ | $8.8 \cdot 10^{-1}$ |
| Three-hidden-layer ReLU Network | $1.4 \cdot 10^{-2}$ | $8.4 \cdot 10^{-4}$ | $-2.1 \cdot 10^0$ | $8.2 \cdot 10^{-1}$ |
| CNN ReLU Network | $3.1 \cdot 10^{-2}$ | $1.4 \cdot 10^{-3}$ | $-1.7 \cdot 10^0$ | $8.7 \cdot 10^{-1}$ |

**CIFAR10** The `CIFAR10` dataset contains $50\,000$ training images and $10\,000$ testing images, both with dimensions of $32 \times 32$ pixels. We run the experiments in intervals of 500 samples (training samples) and explore various neural network architectures, including both ReLU feed-forward networks and CNN ReLU. Figure 4(a) illustrates the result of our analysis for a shallow ReLU neural network with the width of 1000, and Figure 4(b) illustrates the result for a CNN ReLU network.

Table 3: Numerical metrics to compare the strength of two curves $(c_1 + \alpha/\sqrt{n})$ and $(c_2 + \beta/n)$ to model the generalization error in different architectures for the `CIFAR10` dataset.

| | MSE | | R-squared | |
|---|---|---|---|---|
| Architectures | $(c_2 + \beta/n)$ | $(c_1 + \alpha/\sqrt{n})$ | $(c_2 + \beta/n)$ | $(c_1 + \alpha/\sqrt{n})$ |
| Shallow ReLU Network | $7.9 \cdot 10^{-2}$ | $8.1 \cdot 10^{-3}$ | $-3.5 \cdot 10^0$ | $5.3 \cdot 10^{-1}$ |
| CNN ReLU Network | $2.9 \cdot 10^{-1}$ | $3.1 \cdot 10^{-2}$ | $-3.7 \cdot 10^0$ | $4.8 \cdot 10^{-1}$ |

**California housing prices (`CHP`)** The version considered in this study comprises 8 numeric input attributes and a dataset of $20\,640$ samples. These samples were randomly divided into $15\,000$ for the training data and the remaining for the test data. The batch size for the training samples is set to 20. We run the experiments in intervals of 200 samples (training samples).

Table 4: Numerical metrics to compare the strength of two curves $(c_1 + \alpha/\sqrt{n})$ and $(c_2 + \beta/n)$ to model the generalization error of a five-hidden layer network with a uniform width of 300 for the `CHP` dataset.

| Architecture | MSE | | R-squared | |
|---|---|---|---|---|
| | $(c_2 + \beta/n)$ | $(c_1 + \alpha/\sqrt{n})$ | $(c_2 + \beta/n)$ | $(c_1 + \alpha/\sqrt{n})$ |
| Five-hidden-layer ReLU Network | $9.9 \cdot 10^{-2}$ | $2.1 \cdot 10^{-2}$ | $2.1 \cdot 10^{-1}$ | $8.4 \cdot 10^{-1}$ |

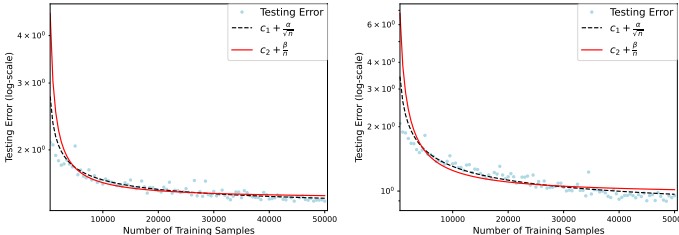

Figure 4: Comparative analysis of the strength of two curves $(c_1 + \alpha/\sqrt{n})$ and $(c_2 + \beta/n)$ to model the generalization error in different architectures for the `CIFAR10` dataset: (a) Shallow ReLU feed-forward network, width of 1000 and (b) CNN ReLU network (left to right).

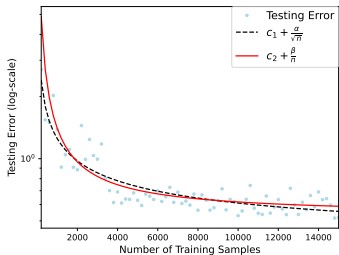

Figure 5: Comparative analysis of the strength of two curves $(c_1 + \alpha/\sqrt{n})$ and $(c_2 + \beta/n)$ to model the generalization error of a five-hidden layer network with a uniform width of 300) for `CHP` dataset

By analyzing both the numerical results and the figures, it is evident that for both **regression and classification** tasks, the generalization error scales more significantly at a rate of $1/\sqrt{n}$ rather than $1/n$. These findings consistently hold true for both ReLU feed-forward networks and ReLU CNNs.

## 6 CONCLUSION

This paper uses Fano's inequality to establish a mini-max risk lower bound for ReLU feed-forward neural networks. The bound scales at the rate $\sqrt{\log(d)/n}$. Our empirical findings support this conclusion and indicate that for both regression and classification problems, the generalization error of ReLU neural networks scales at the rate $1/\sqrt{n}$. More specifically, our theories and empirical results demonstrate that the generalization error of a ReLU feed-forward network cannot be improved beyond the rate $1/\sqrt{n}$ in general. Our theory is also much closer to current practice than earlier works: for example, we focus on deep networks rather than shallow ones (Du et al., 2018; Klusowski & Barron, 2017), and we employ ReLU activation functions instead of Sinusoidal (Klusowski & Barron, 2017) or linear (Du et al., 2018) activation functions.

The current discussions about large language models (LLMs) illustrate that our research is extremely relevant and timely. The training data for LLMs have steadily grown over the last years: for example, GPT-3 was trained on about 300 billion tokens (Brown et al., 2020), Chinchilla on about 1.4 trillion tokens (Hoffmann et al., 2022), GPT-4 on about 13 trillion tokens (OpenAI, 2023) and Llama 3 on over 15 trillion tokens (Meta, 2024). However, it is believed that this trend will have a natural end soon: there will simply not be enough fresh texts any more. A trend is, therefore, the development of smaller models, such as Phi-3-Mini (Abdin et al., 2024), which are trained on less data in general and on "real" data enriched with synthetic data. But how much data are really needed for such minimal AI-solutions? This paper is a small step toward answering this question.

## 7 ACKNOWLEDGMENTS

This research was partially funded by grants 50906238, 01IS24065B, 543964668 (SPP2298), and 520388526 (TRR391) from the Deutsche Forschungsgemeinschaft (DFG, German Research Foundation).

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

## A   FURTHER TECHNICAL RESULTS

In this section, we present additional technical results from the work of others and our own, that are essential for the proof of Theorem 2.2's components but might also be of interest by themselves. We divide the results into two main parts. The first part includes Lemma A.5 and a few other results that are contained in the proof of this lemma, and the second part includes a few results, both from our work and others' to prove Lemma 3.2. Before providing those preliminary results, we provide the general formula for Fano's inequality (Wainwright, 2019, Proposition 15.12). Based on the concept of packing number, assume that $\{\mathbb{P}^n_{f_{\Theta^1}}, \dots, \mathbb{P}^n_{f_{\Theta^{\mathcal{M}}}}\}$ is a family of $n$-product distributions (as defined in Part 1) for the corresponding neural networks $f_{\Theta^1}, \cdots, f_{\Theta^{\mathcal{M}}}$ which satisfy $\rho(f_{\Theta^j}(\boldsymbol{x}), f_{\Theta^k}(\boldsymbol{x})) \geq 2\delta$ for all $j, k \in [\mathcal{M}]$ and $j \neq k$. Then, assume that $J$ is uniformly distributed over the index set $[\mathcal{M}]$ and the conditional distribution of $(Y^n|X^n)$ given $J$ defined by $((Y^n|X^n) \mid J = j) \sim \mathbb{P}^n_{f_{\Theta^j}}$. Accordingly, Fano's inequality can be formalized as:

**Lemma A.1 (Fano's Inequality)** *Let $\{f_{\Theta^1}, \dots, f_{\Theta^{\mathcal{M}}}\} \subseteq \mathcal{F}_{\mathcal{B}_{\mathrm{L}}}$ be a $2\delta$-packing set with respect to $\rho$. Then, for any increasing function $\Phi : [0, \infty) \to [0, \infty)$, the mini-max risk is lower bounded by*

$$\mathcal{R}_{(n,d)}(\mathcal{F}_{\mathcal{B}_{\mathrm{L}}}; \Phi \circ \rho) \geq \Phi(\delta)\left(1 - \frac{I(J; Y^n|X^n) + \log 2}{\log \mathcal{M}}\right).$$

The symbol $I(J; Y^n|X^n)$ represents the mutual information between a random index $J$, which is drawn uniformly from the index set $[\mathcal{M}]$ and the samples $(Y^n|X^n)$ drawn from the prior distribution $\mathbb{P}^n_{f_{\Theta^j}}$ corresponding to $f_{\Theta^j} := f_{\Theta^J}$. The mutual information, measures how much information can be revealed about the index $J$ of a $2\delta$-packing set by drawing the samples $(Y^n|X^n)$.

PART 1: PRELIMINARY RESULTS FOR UPPER BOUNDING THE MUTUAL INFORMATION

We present some auxiliary results that are contained in the proof of Lemma A.5. To follow these results more conveniently, we explain the necessary steps briefly. After defining the KL divergence as a measure of distance between two probability measures, we calculate the KL divergence between two multivariate normal distributions Hayakawa & Suzuki (2020, Lemma A.1). Then, we calculate KL divergence of $n$-product of two multivariate normal distributions and finally, we find the connection between the mutual information and KL divergence.

The KL divergence between two different probability distributions $\mathbb{P}$ and $\mathbb{Q}$ on domain $\mathcal{X}$ with densities $p(\boldsymbol{x})$ and $q(\boldsymbol{x})$, respectively, can be defined as (Wainwright, 2019, Equation 3.57)

$$D_{\mathrm{KL}}\big(\mathbb{P} \parallel \mathbb{Q}\big) = \int_{\boldsymbol{x} \in \mathcal{X}} p(\boldsymbol{x}) \log \frac{p(\boldsymbol{x})}{q(\boldsymbol{x})} d\boldsymbol{x}.$$

Since we deal with $n$ independent samples, the probabilities are defined on a product space of $n$ components. Therefore, we need to find the KL divergence between two different $n$-product distributions. Assume that $(\mathbb{P}^1, \dots, \mathbb{P}^n)$ be a collection of $n$ probability distributions, and define $\mathbb{P}^{1:n} := \bigotimes_{i=1}^n \mathbb{P}^i$ as the $n$-product distributions. Define another $n$-product distribution $\mathbb{Q}^{1:n}$ in a similar way. For the ease of notation, we define $\mathbb{P}^n := \mathbb{P}^{1:n}$ and $\mathbb{Q}^n := \mathbb{Q}^{1:n}$. Then, the connection between the KL divergence of $n$-product distributions $\mathbb{P}^n$ and $\mathbb{Q}^n$ and the KL divergence of the individual pairs (Wainwright, 2019, Equation 15.11a), can be formalized as the following lemma:

**Lemma A.2 (Decomposition of the** KL divergence **for $n$-product distributions)** *For two $n$-product distributions $\mathbb{P}^n$ and $\mathbb{Q}^n$, it holds that*

$$D_{\mathrm{KL}}\big(\mathbb{P}^n \parallel \mathbb{Q}^n\big) = \sum_{i=1}^n D_{\mathrm{KL}}\big(\mathbb{P}^i \parallel \mathbb{Q}^i\big).$$

*And in the case of i·i·d· product distributions — meaning that $P^i = P^1$ and $\mathbb{Q}^i = \mathbb{Q}^1$ for all $i \in \{1, \dots, n\}$— we have*

$$D_{\mathrm{KL}}\big(\mathbb{P}^n \parallel \mathbb{Q}^n\big) = n \times D_{\mathrm{KL}}\big(\mathbb{P}^1 \parallel \mathbb{Q}^1\big).$$

*We consider short-hands $\mathbb{P}$ and $\mathbb{Q}$, for $\mathbb{P}^1$ and $\mathbb{Q}^1$, respectively. So, the previous equation takes form*

$$D_{\mathrm{KL}}\big(\mathbb{P}^n \parallel \mathbb{Q}^n\big) = n \times D_{\mathrm{KL}}\big(\mathbb{P} \parallel \mathbb{Q}\big).$$

We then proceed to calculate the KL divergence between two normal distributions. Consider the regression model defined in Equation (1) and the network model defined in Equation (2). We assume that the noise terms are independent and identically distributed and $u_i \in \mathcal{N}(0, \sigma^2)$. Recall that the explanatory variables $\boldsymbol{x}_i$ follow a fixed distribution $\mathbb{P}_{\boldsymbol{x}}$ and have the density $h(\boldsymbol{x})$. Then, we define $\boldsymbol{z} := (\boldsymbol{x}, y) \in \mathbb{R}^d \times \mathbb{R}$ as the joint variable of $\boldsymbol{x}$ and $y$. According to the conditional probability, the joint density can be written as follows:

$$
\begin{aligned}
p_{f_{\boldsymbol{\Theta}^j}}(\boldsymbol{z}) &= p_{Y|X}(y|\boldsymbol{x})h(\boldsymbol{x}) \\
&= \frac{1}{\sqrt{2\pi\sigma^2}} e^{-\frac{\left(y - f_{\boldsymbol{\Theta}^j}(\boldsymbol{x})\right)^2}{2\sigma^2}} h(\boldsymbol{x}),
\end{aligned}
\tag{7}
$$

where $j \in [\mathcal{M}]$ and $p_{f_{\boldsymbol{\Theta}^j}}(\boldsymbol{z})$ is the joint density of $(\boldsymbol{x}, y)$ with regression function $f_{\boldsymbol{\Theta}^j}(\boldsymbol{x})$. And consider $p_{f_{\boldsymbol{\Theta}^k}}(\boldsymbol{z})$ as another joint density of $(\boldsymbol{x}, y)$ in the same manner with regression function $f_{\boldsymbol{\Theta}^k}(\boldsymbol{x})$ and two distinct corresponding normal distributions $\mathbb{P}_{f_{\boldsymbol{\Theta}^j}}$ and $\mathbb{P}_{f_{\boldsymbol{\Theta}^k}}$ such that have densities $p_{f_{\boldsymbol{\Theta}^j}}(\boldsymbol{z})$ and $p_{f_{\boldsymbol{\Theta}^k}}(\boldsymbol{z})$, respectively. Recall that $f_{\boldsymbol{\Theta}^j}$ and $f_{\boldsymbol{\Theta}^k}$ are any two distinct neural networks of the neural network model defined in Section 2, which parameterized by $\boldsymbol{\Theta}^j$ and $\boldsymbol{\Theta}^k$ ($j, k \in [\mathcal{M}]$ as any two distinct indices of the $2\delta$-packing set). Then, the KL divergence between any two normal distributions $\mathbb{P}_{f_{\boldsymbol{\Theta}^j}}$ and $\mathbb{P}_{f_{\boldsymbol{\Theta}^k}}$ can be calculated as the following lemma (Yang & Barron, 1999):

**Lemma A.3 (The** KL divergence **between two multivariate normal distributions)** *Assume any two normal distributions $\mathbb{P}_{f_{\boldsymbol{\Theta}^j}}$ and $\mathbb{P}_{f_{\boldsymbol{\Theta}^k}}$ for all $j, k \in [\mathcal{M}]$ and $j \neq k$, then it holds that*

$$
D_{\mathrm{KL}}(\mathbb{P}_{f_{\boldsymbol{\Theta}^j}} \parallel \mathbb{P}_{f_{\boldsymbol{\Theta}^k}}) = \frac{1}{2\sigma^2} \int_{\boldsymbol{x} \in \mathcal{X}} \left(f_{\boldsymbol{\Theta}^j}(\boldsymbol{x}) - f_{\boldsymbol{\Theta}^k}(\boldsymbol{x})\right)^2 h(\boldsymbol{x}) d\boldsymbol{x}
$$

*And that*

$$
D_{\mathrm{KL}}(\mathbb{P}_{f_{\boldsymbol{\Theta}^j}}^n \parallel \mathbb{P}_{f_{\boldsymbol{\Theta}^k}}^n) = \frac{n}{2\sigma^2} \int_{\boldsymbol{x} \in \mathcal{X}} \left(f_{\boldsymbol{\Theta}^j}(\boldsymbol{x}) - f_{\boldsymbol{\Theta}^k}(\boldsymbol{x})\right)^2 h(\boldsymbol{x}) d\boldsymbol{x}.
$$

To fulfill this part's goal, we have just left to find a connection between the KL divergence and the mutual information.

In the next lemma, we are interested to upper bounding the mutual information (Scarlett & Cevher, 2021) —which measures the dependence between the joint distributions and the product of the marginals of two random variables— by describing it's connection with the KL divergence. Assume that under the Markov chain $J \to f_{\boldsymbol{\Theta}^J} \to (Y^n|X^n)$, a random index $J$ is drawn uniformly from $\{1, \ldots, \mathcal{M}\}$ and samples $(Y^n|X^n)$ are drawn from the prior distributions $\mathbb{P}_{f_{\boldsymbol{\Theta}^j}}^n$ corresponding to $f_{\boldsymbol{\Theta}^j} := f_{\boldsymbol{\Theta}^J}$. Note that if one sample $(Y|X)$ drawn, then we have $I(J; Y|X)$.

There are many tools to upper bounding the mutual information and the most straight forward tools is based on the KL divergence (Wainwright, 2019, Equation 15.34) as follows:

**Lemma A.4 (The connection between the mutual information and the** KL divergence**)** *For any two distinct normal probability distributions $\mathbb{P}_{f_{\boldsymbol{\Theta}^j}}$ and $\mathbb{P}_{f_{\boldsymbol{\Theta}^k}}$ for all $j, k \in [\mathcal{M}]$, it holds that*

$$
I(J; Y|X) \leq \frac{1}{\mathcal{M}^2} \sum_{\substack{j,k=1 \\ j \neq k}}^{\mathcal{M}} D_{\mathrm{KL}}(\mathbb{P}_{f_{\boldsymbol{\Theta}^j}} \parallel \mathbb{P}_{f_{\boldsymbol{\Theta}^k}}).
$$

*For any two distinct $n$-product normal probability distributions $\mathbb{P}_{f_{\boldsymbol{\Theta}^j}}^n$ and $\mathbb{P}_{f_{\boldsymbol{\Theta}^k}}^n$, it holds that*

$$
I(J; Y^n|X^n) \leq \frac{n}{\mathcal{M}^2} \sum_{\substack{j,k=1 \\ j \neq k}}^{\mathcal{M}} D_{\mathrm{KL}}(\mathbb{P}_{f_{\boldsymbol{\Theta}^j}} \parallel \mathbb{P}_{f_{\boldsymbol{\Theta}^k}}).
$$

**Lemma A.5 (Upper bounding $I(J; Y^n|X^n)$ of the $2\delta$-packing of network's space $\mathcal{F}_{\mathcal{B}_{\mathrm{L}}}$)** *For all possible pairs of two distinct networks $f_{\boldsymbol{\Theta}^j}, f_{\boldsymbol{\Theta}^k} \in \mathcal{F}_{\mathcal{B}_{\mathrm{L}}}$ satisfy $\rho(f_{\boldsymbol{\Theta}^j}(\boldsymbol{x}), f_{\boldsymbol{\Theta}^k}(\boldsymbol{x})) \geq 2\delta$, the mutual information $I(J; Y^n|X^n)$ is upper bounded by*

$$
I(J; Y^n|X^n) \leq \frac{2n(\kappa\delta)^2}{\sigma^2},
$$

*for a suitable* $\kappa \in [1, \infty)$, *such that* $\rho(f_{\boldsymbol{\Theta}^j}(\boldsymbol{x}), f_{\boldsymbol{\Theta}^k}(\boldsymbol{x})) \leq 2\kappa\delta$.

PART 2: RELATIONSHIP BETWEEN GLOBAL AND LOCAL METRIC ENTROPIES

Here, we briefly provide a connection between the global metric entropy and the local metric entropy (Yang & Barron, 1999, Section 7). In fact, the global metric entropy ensures the existence of "at least one" local packing set which has the property required for the use of Birgé's argument (Birgé, 1983). Accordingly, in the proof of our main Theorem 2.2, instead of considering the entire space $(\mathcal{F}_{\mathcal{B}_L})$, we focus on a local area (ball with radius $2\kappa\delta$) where the "local" Fano's method is applied. The following Definition A.6 (Yang & Barron, 1999, Section 7) and Lemma A.7 (Yang & Barron, 1999, Lemma 3) assist us in determining the value for $\kappa$, ensuring that there is no concern about $\kappa$ being too large. These also allow us to work with a more manageable subset of the function space $\mathcal{F}_{\mathcal{B}_L}$, called $\mathcal{F}_S$ in the proof of our Theorem 2.2. By considering the case when $d$ is a metric and assuming the global packing entropy of the space $S$ under the distance $d$ is $M(\delta)$, the definition of the local metric entropy is as follows (Yang & Barron, 1999, Section 7):

**Definition A.6 (Local metric entropy)** *The local $\delta$-entropy at $\theta \in S$ is the log of the largest $(\delta/2)$-packing set in $B(\theta, \delta) = \{\theta' \in S : d(\theta', \theta) \leq \delta\}$. The local $\delta$-entropy at $\theta$ is denoted by $M(\delta|\theta)$ and the local $\delta$-entropy of $S$ is defined as $M^{loc}(\delta) = \max_{\theta \in S} M(\delta|\theta)$.*

Based on this definition, the relationship between the global and local entropies can be formulate as follows (Yang & Barron, 1999, Lemma 3):

**Lemma A.7 (The relationship between the global and local metric entropies)** *the global and local metric entropies have the following relationship:*

$$M(\delta/2) - M(\delta) \leq M^{loc}(\delta) \leq M(\delta/2).$$

**Value of $\kappa$ :** Employing Lemma A.7, $\log \mathcal{M}\big(2\delta, \mathcal{F}_S, \|\cdot\|_{L_2}\big)$—that is a local entropy— can be lower bounded on a high-level by a fraction of $\log \mathcal{M}\big(2\delta, \mathcal{F}_{\mathcal{B}_L}, \|\cdot\|_{L_2}\big)$, that is a global entropy (Lemma 3.2). In the proof of Theorem 2.2, we used the result of Lemma A.7 to bound the local packing. We proceed as follows: we begin with $2\delta$-packing, meaning that $\rho(f_{\boldsymbol{\Theta}^j}(\boldsymbol{x}), f_{\boldsymbol{\Theta}^k}(\boldsymbol{x})) \geq 2\delta$ for all $j \neq k \in [\mathcal{M}]$. It is easy to conclude that these centers will be within $4\delta$ from $\theta$. By applying the triangle inequality, we conclude that $\rho(f_{\boldsymbol{\Theta}^j}(\boldsymbol{x}), f_{\boldsymbol{\Theta}^k}(\boldsymbol{x})) \leq 8\delta$. As we have $\rho(f_{\boldsymbol{\Theta}^j}(\boldsymbol{x}), f_{\boldsymbol{\Theta}^k}(\boldsymbol{x})) \leq 2\kappa\delta$ (see Lemma A.5), this gives us $\kappa = 4$. We use this value directly in the proof of Theorem 2.2.

PART 3: PRELIMINARY RESULTS FOR DERIVING A LOWER BOUND FOR PACKING NUMBER OF ReLU NETWORKS' SPACES

In this section, we present supporting lemmas that are included in the proof of Lemma 3.2 for deriving the lower bound for the packing number of our defined ReLU network's space. We start by calculating the Gaussian integrals over a half-space. Assume that $\boldsymbol{x}$ is a realization of random variable $X$ that follows the $d$-dimensional Gaussian distribution, then we say that for $k \in \{1, \ldots S\}$, $\boldsymbol{b}_k^\top \boldsymbol{x} > 0$ and $\boldsymbol{b}_k^\top \boldsymbol{x} \leq 0$ are two half-spaces of hyperplane $\boldsymbol{b}_k^\top \boldsymbol{x} = 0$ for $\boldsymbol{b}_k \in \mathbb{R}^d$. We then can define the probability density function of $\boldsymbol{x}$ with mean vector $\boldsymbol{\mu}$ and the covariance matrix $\boldsymbol{\Sigma}$ as follows:

$$p(\boldsymbol{x}, \boldsymbol{\mu}, \boldsymbol{\Sigma}) = \frac{1}{(2\pi)^{d/2}\sqrt{|\boldsymbol{\Sigma}|}} \int_{\boldsymbol{x} \in \mathcal{X}} e^{\frac{-(\boldsymbol{x}-\boldsymbol{\mu})^\top \boldsymbol{\Sigma}^{-1}(\boldsymbol{x}-\boldsymbol{\mu})}{2}} d\boldsymbol{x},$$

where $|\boldsymbol{\Sigma}| \equiv \det(\boldsymbol{\Sigma})$, is the determinant of $\boldsymbol{\Sigma}$.

If $\boldsymbol{\mu} = \boldsymbol{0}$, then we have

$$p(\boldsymbol{x}, \boldsymbol{0}, \boldsymbol{\Sigma}) = \frac{1}{(2\pi)^{d/2}\sqrt{|\boldsymbol{\Sigma}|}} \int_{\boldsymbol{x} \in \mathcal{X}} e^{\frac{-\boldsymbol{x}^\top \boldsymbol{\Sigma}^{-1}\boldsymbol{x}}{2}} d\boldsymbol{x}.$$

Accordingly, the probability density function of $\boldsymbol{x}$ on either half-space $\boldsymbol{b}_k^\top \boldsymbol{x} > 0$ or $\boldsymbol{b}_k^\top \boldsymbol{x} \leq 0$ takes the form

$$p(\boldsymbol{b}_k^\top \boldsymbol{x} > 0, \boldsymbol{0}, \boldsymbol{\Sigma}) = \frac{1}{(2\pi)^{d/2}\sqrt{|\boldsymbol{\Sigma}|}} \int_{\boldsymbol{b}_k^\top \boldsymbol{x} > \boldsymbol{0}} e^{\frac{-\boldsymbol{x}^\top \boldsymbol{\Sigma}^{-1}\boldsymbol{x}}{2}} d\boldsymbol{x},$$

and can be calculated as the following lemma:

**Lemma A.8 (Gaussian integrals over a half-space)** *Assume that $\boldsymbol{x}, \boldsymbol{b}_k \in \mathbb{R}^d$ and for a fixed vector $\boldsymbol{b}_k$, we define a half-space $\boldsymbol{b}_k^\top \boldsymbol{x} > 0$. Then, for the corresponding probability density function it holds that*

$$p(\boldsymbol{b}_k^\top \boldsymbol{x} > 0, \boldsymbol{0}, \boldsymbol{\Sigma}) = \frac{1}{2}\,.$$

In the next lemma we are motivated to employ the result of Lemma A.8 to calculate $\mathbb{E}[(\phi(\boldsymbol{b}_k^\top \boldsymbol{x}))^2]$ which is necessary for the proof of Lemma 3.2.

**Lemma A.9 (The expectation of squared ReLU functions)** *Let $\boldsymbol{x}$ be a Gaussian random variable and $\boldsymbol{b}_k^\top \boldsymbol{x} > 0$ is a half-space, then,*

$$\mathbb{E}\left[\left(\phi(\boldsymbol{b}_k^\top \boldsymbol{x})\right)^2\right] = \frac{1}{2}\,.$$

In the following lemma, we present Klusowski & Barron (2017, Lemma1), which concerns the cardinality of a set and is integral to the proof of Lemma 3.2. This lemma helps us define our desired set with a predefined Hamming weight, and its elements can be interpreted as binary codes. Now, let state the lemma.

**Lemma A.10 (Cardinality of a binary set)** *For integers $d$ and $d'$ with $d \in [10, \infty)$ and $d' \in [1, d/10]$, define a set*

$$\mathcal{S} := \left\{\boldsymbol{w} \in \{0, 1\}^d : \|\boldsymbol{w}\|_1 = d'\right\}.$$

*Then, there exists a subset $\mathcal{A} \subset \mathcal{S}$ with cardinality at least $S := \sqrt{\binom{d}{d'}}$ such that each element has Hamming weight $d'$ and any pairs of elements have minimum Hamming distance $d'/5$.*

# B  PROOFS

In this section, we provide the proof of Lemma 3.2. Additionally, the proofs of the lemmas in Appendix A will be included.

## B.1  PROOF OF LEMMA 3.2

**Proof**  The core of this proof involves two steps: first, the construction of a subclass of functions within $\mathcal{F}_{\mathcal{B}_L}$, and then finding a lower bound for $\log$ of the cardinality of the constructed class. Second, using the fact that a lower bound for the cardinality of a smaller space can serve as a lower bound for the cardinality of the larger space. Let us begin by discussing the construction of the subclass of the function class $\mathcal{F}_{\mathcal{B}_L}$.

STEP 1: THE CONSTRUCTION A SUBCLASS OF FUNCTION CLASS $\mathcal{F}_{\mathcal{B}_L}$

Our first step is to construct a subclass of our defined function class $\mathcal{F}_{\mathcal{B}_L}$ and then to find a lower bound for the $\log$ of the packing number of the constructed class. To achieve this, we begin by defining a set of binary vectors $\mathcal{C} \in \{0, 1\}^d$ for $d \in [10, \infty)$ such that each element of this set has a Hamming weight of $d'$, where $d' \in [1, d/10]$ and the cardinality of this set is denoted by $S$. Recall that, we assume that $v_0 = 1$, so, we can choose $d' = v_0^2 = 1$. Then, anyone can readily conclude that $S = d$ and we can consider the vector $\boldsymbol{b}_i \in \{0, 1\}^d$ as a vector with all the entries equal to zero except for the $i$th entry, which is set to one. It implies that for all $i \neq j \in \{1, \ldots, S\}$

$$|\boldsymbol{b}_i^\top \boldsymbol{b}_j| = 0\,.$$

We can also conclude that for each $\boldsymbol{b}_i$ with $i \in \{1, \ldots, S\}$, we have

$$\|\boldsymbol{b}_i\|_2 = \sqrt{\left((\boldsymbol{b}_i)_1\right)^2 + \ldots + \left((\boldsymbol{b}_i)_d\right)^2} = 1\,.$$

Following the same argument as above we have

$$\|\boldsymbol{b}_i\|_1 = \left|(\boldsymbol{b}_i)_1\right| + \ldots + \left|(\boldsymbol{b}_i)_d\right| = 1\,.$$

Then, for an enumeration $\boldsymbol{b}_1, \ldots, \boldsymbol{b}_S$ of $\mathcal{C}$, we claim that the function space $\mathcal{F}_0$ as defined below can represent a subspace of $\mathcal{F}_{\mathcal{B}_{\mathrm{L}}}$:

$$\mathcal{F}_0 := \left\{ f_{(\overline{\overline{\boldsymbol{w}}}, \boldsymbol{b})}(\boldsymbol{x}) := \frac{V_{\mathcal{F}}}{\lambda} \sum_{k=1}^{S} \overline{\overline{w}}_k \phi^1(\boldsymbol{b}_k^\top \boldsymbol{x}) \ : \overline{\overline{\boldsymbol{w}}} \in \overline{\overline{\mathcal{A}}} \right\}$$

with $\overline{\overline{\boldsymbol{w}}} := (V_{\mathcal{F}}/\lambda)\overline{\boldsymbol{w}}$ and $\overline{\overline{\mathcal{A}}} := \{\overline{\overline{\boldsymbol{w}}} \in \{0, \tau\}^S \ : \|\overline{\overline{\boldsymbol{w}}}\|_1 = \tau\lambda\}$ for $V_{\mathcal{F}}, \tau \in (0, \infty)$ and an integer $\lambda \in [1, S/10]$ (exact values of $V_{\mathcal{F}}$ and $\lambda$ be specified later in the proof, while the $\tau$ is a constant depending on $v_{\mathrm{s}}$.) We argue that $\mathcal{F}_0$ is representing a subspace of deep neural networks $\mathcal{F}_{\mathcal{B}_{\mathrm{L}}}$ with non-negative weight matrices $W^l$ for $l \in \{1, \ldots, L\}$ inspired by Hebiri et al. (2025). To be more precise, we state that for a fixed pair $\lambda, \tau$ and $\forall \, \overline{\overline{\boldsymbol{w}}} \in \overline{\overline{\mathcal{A}}}$, there exists a tuple of non-negative weight matrices $(W^1, \ldots, W^L)$ with $\sum_{l=1}^{L} \|W^l\|_1 \leq v_{\mathrm{s}}$ that their product verifies $W^L W^{L-1} \ldots W^1 = \overline{\overline{\boldsymbol{w}}}$ (see Remark B.1). For example, for a large enough $v_{\mathrm{s}}$, we can fix $\tau = 1$, since our network space $\mathcal{F}_{\mathcal{B}_{\mathrm{L}}}$ can generate binary vectors $\overline{\overline{\boldsymbol{w}}}$ with cardinality $S$ employing a suitable tuple of non-negative matrices $(W^1, \ldots, W^L)$. In fact, we are using the property that for non-negative weight matrices, the ReLU activations in deep neural networks (for layers $l \in \{1, \ldots, L\}$) can be ignored and so, deep network for the corresponding layers behaves like a simple matrix product (Hebiri et al., 2025). This property gives us the chance to write those specific networks (with non-negative weight matrices) in the form of a shallow neural network as stated in $\mathcal{F}_0$. Let's also note that for the corresponding networks, we can 1. invoke compatible norms $L$ times, 2. use the inequality of arithmetic and geometric means, 3. definition of our ReLU network's space (defined in Section 2) and 4. invoke the definition of $V_{\mathcal{F}}$—in the view of Theorem 2.2—, to get

$$\|W^L W^{L-1} \cdots W^1\|_1 \leq \|W^L\|_1 \|W^{L-1} \cdots W^1\|_\infty$$

$$\vdots$$

$$\leq \|W^L\|_1 \|W^{L-1}\|_1 \|W^{L-2}\|_1 \cdots \|W^1\|_1$$

$$\leq \left( \frac{1}{L} \sum_{i=1}^{L} \|W^i\|_1 \right)^L$$

$$\leq \left( \frac{v_{\mathrm{s}}}{L} \right)^L$$

$$= V_{\mathcal{F}}.$$

Some rewriting over the $\mathcal{F}_0$ implies

$$\mathcal{F}_0 = \left\{ f_{(\overline{\boldsymbol{w}}, \boldsymbol{b})}(\boldsymbol{x}) = \frac{\tau V_{\mathcal{F}}}{\lambda} \sum_{k=1}^{S} w_k \phi^1(\boldsymbol{b}_k^\top \boldsymbol{x}) \ : \boldsymbol{w} \in \mathcal{A} \right\},$$

where $\overline{\boldsymbol{w}} := (\tau V_{\mathcal{F}}/\lambda)\boldsymbol{w}$ and $\mathcal{A} := \{\boldsymbol{w} \in \{0, 1\}^S \ : \|\boldsymbol{w}\|_1 = \lambda\}$ is the set in Lemma A.10. $\lambda \in [1, S/10]$ is an integer and denotes as the Hamming weight of each element of this set (Klusowski & Barron, 2017, Theorem 2).

According to the above definition of $\mathcal{F}_0$, we have

$$\mathbb{E}\left[\|f_{(\overline{\boldsymbol{w}}, \boldsymbol{b})}(\boldsymbol{x}) - f_{(\overline{\boldsymbol{w}}', \boldsymbol{b})}(\boldsymbol{x})\|_{L_2}^2\right] = \left( \frac{\tau V_{\mathcal{F}}}{\lambda} \right)^2 \mathbb{E}\left[ \left( \sum_{k=1}^{S} (w_k - w_k') \phi^1(\boldsymbol{b}_k^\top \boldsymbol{x}) \right)^2 \right],$$

where $\boldsymbol{w}, \boldsymbol{w}' \in \mathcal{A}$.

Note that, based on the structure of $\boldsymbol{w}$ and $\boldsymbol{w}'$, for all $k \in \{1, \ldots, S\}$, $(w_k - w_k')$ falls within the set $\{-1, 0, 1\}$. And if $(w_k - w_k') = 0$, the value of the expected term on the right-hand side –for the corresponding $k$– is equal to 0; thus for the sake of convenience, we consider an integer value $S' < S$ in such a way that $|w_k - w_k'| = 1$ for all $k \in \{1, \ldots, S'\}$. Based on the structure of all pairs $\boldsymbol{w}, \boldsymbol{w}' \in \mathcal{A}$ and Lemma A.10, we can conclude that $S' \geq \lambda/5$. We will use $S'$ for the remainder of the proof.

We then proceed with

$$\mathbb{E}\left[\|f_{(\overline{\boldsymbol{w}}, \boldsymbol{b})}(\boldsymbol{x}) - f_{(\overline{\boldsymbol{w}}', \boldsymbol{b})}(\boldsymbol{x})\|_{L_2}^2\right] = \left( \frac{\tau V_{\mathcal{F}}}{\lambda} \right)^2 \mathbb{E}\left[ \left( \sum_{k=1}^{S'} ((w_k - w_k') \phi^1(\boldsymbol{b}_k^\top \boldsymbol{x})) \right)^2 \right].$$

Next, we are motivated to find a lower bound for $\mathbb{E}[\|f_{(\overline{\boldsymbol{w}},\boldsymbol{b})}(\boldsymbol{x}) - f_{(\overline{\boldsymbol{w}}',\boldsymbol{b})}(\boldsymbol{x})\|^2_{L_2}]$. We can 1. employ the last view, 2. expand the squared over the sum and invoke the linearity of expected value, 3. invoke the linearity of expected value, 4. apply the above assumption that $(w_k - w_k')^2 = 1$, 5. use the result of Lemma A.9, which shows $\mathbb{E}[(\phi(\boldsymbol{b}_k^\top \boldsymbol{x}))^2] = 1/2$, and the properties of sum function and 6. use the fact that $\mathbb{E}[\phi^1(\boldsymbol{b}_k^\top \boldsymbol{x})\phi^1(\boldsymbol{b}_j^\top \boldsymbol{x})] = 1/2\pi$, and the properties of sum function to obtain

$$
\begin{aligned}
\mathbb{E}\big[\|f_{(\overline{\boldsymbol{w}},\boldsymbol{b})}(\boldsymbol{x}) - f_{(\overline{\boldsymbol{w}}',\boldsymbol{b})}(\boldsymbol{x})\|^2_{L_2}\big] &= \Big(\frac{\tau V_\mathcal{F}}{\lambda}\Big)^2 \mathbb{E}\bigg[\Big(\sum_{k=1}^{S'}\big((w_k - w_k')\phi^1(\boldsymbol{b}_k^\top \boldsymbol{x})\big)\Big)^2\bigg] \\
&= \Big(\frac{\tau V_\mathcal{F}}{\lambda}\Big)^2 \bigg(\sum_{k=1}^{S'}\mathbb{E}\Big[\big((w_k - w_k')\phi^1(\boldsymbol{b}_k^\top \boldsymbol{x})\big)^2\Big] \\
&\qquad + \mathbb{E}\bigg[\sum_{\substack{k=1 \\ }}^{S'}\sum_{\substack{j=1 \\ j\neq k}}^{S'}(w_k - w_k')\phi^1(\boldsymbol{b}_k^\top \boldsymbol{x})(w_j - w_j')\phi^1(\boldsymbol{b}_j^\top \boldsymbol{x})\bigg]\bigg) \\
&= \Big(\frac{\tau V_\mathcal{F}}{\lambda}\Big)^2 \bigg(\sum_{k=1}^{S'}\mathbb{E}\Big[\big((w_k - w_k')\phi^1(\boldsymbol{b}_k^\top \boldsymbol{x})\big)^2\Big] \\
&\qquad + \sum_{\substack{k=1 \\ }}^{S'}\sum_{\substack{j=1 \\ j\neq k}}^{S'}\mathbb{E}\big[(w_k - w_k')\phi^1(\boldsymbol{b}_k^\top \boldsymbol{x})(w_j - w_j')\phi^1(\boldsymbol{b}_j^\top \boldsymbol{x})\big]\bigg) \\
&= \Big(\frac{\tau V_\mathcal{F}}{\lambda}\Big)^2 \bigg(\sum_{k=1}^{S'}\mathbb{E}\Big[\big(\phi^1(\boldsymbol{b}_k^\top \boldsymbol{x})\big)^2\Big] \\
&\qquad + \sum_{\substack{k=1 \\ }}^{S'}\sum_{\substack{j=1 \\ j\neq k}}^{S'}\mathbb{E}\big[(w_k - w_k')\phi^1(\boldsymbol{b}_k^\top \boldsymbol{x})(w_j - w_j')\phi^1(\boldsymbol{b}_j^\top \boldsymbol{x})\big]\bigg) \\
&= \Big(\frac{\tau V_\mathcal{F}}{\lambda}\Big)^2 \bigg(\frac{S'}{2} + \sum_{\substack{k=1 \\ }}^{S'}\sum_{\substack{j=1 \\ j\neq k}}^{S'}(w_k - w_k')(w_j - w_j')\mathbb{E}\big[\phi^1(\boldsymbol{b}_k^\top \boldsymbol{x})\phi^1(\boldsymbol{b}_j^\top \boldsymbol{x})\big]\bigg) \\
&= \Big(\frac{\tau V_\mathcal{F}}{\lambda}\Big)^2 \bigg(\frac{S'}{2} + \frac{1}{2\pi}\sum_{\substack{k=1 \\ }}^{S'}\sum_{\substack{j=1 \\ j\neq k}}^{S'}(w_k - w_k')(w_j - w_j')\bigg),
\end{aligned}
$$

here, we need to analyse the second term. We know that $|w_k - w_k'| = 1$. Thus, $(w_k - w_k')$, $(w_j - w_j')$ for $k, j \in \{1, \ldots, S'\}$ can have either the same signs or different signs that result in the corresponding positive or negative terms. The worst case happens when the number of $+1$ and $-1$ cases are equal. There, the number of total terms is $S'(S' - 1)$, and we have $S'((S'/2) - 1)$ number of positive terms and the remaining terms are negative ($S'^2/2$). So, we can proceed the lower bounding by 1. employing the last view, 2. considering the worst case scenario as discussed, 3. performing some simplifications, 4. performing some arithmetic math, 5. using the fact that $(\pi - 1)/2\pi) > 2/10$,

6. applying the conclusion that $S' \geq \lambda/5$ and 7. performing some simplifications to obtain

$$\mathbb{E}\big[\|f_{(\overline{\boldsymbol{w}},\boldsymbol{b})}(\boldsymbol{x}) - f_{(\overline{\boldsymbol{w}}',\boldsymbol{b})}(\boldsymbol{x})\|^2_{L_2}\big] = \Big(\frac{\tau V_{\mathcal{F}}}{\lambda}\Big)^2 \Big(\frac{S'}{2} + \frac{1}{2\pi}\sum_{k=1}^{S'}\sum_{\substack{j=1\\j\neq k}}^{S'}(w_k - w'_k)(w_j - w'_j)\Big)$$

$$\geq \Big(\frac{\tau V_{\mathcal{F}}}{\lambda}\Big)^2 \Big(\frac{S'}{2} + \frac{1}{2\pi}\Big(S'\Big(\frac{S'}{2} - 1\Big) - \frac{S'^2}{2}\Big)\Big)$$

$$= \Big(\frac{\tau V_{\mathcal{F}}}{\lambda}\Big)^2 \Big(\frac{S'}{2} + \frac{-S'}{2\pi}\Big)$$

$$= \Big(\frac{\tau V_{\mathcal{F}}}{\lambda}\Big)^2 \Big(S'\Big(\frac{\pi - 1}{2\pi}\Big)\Big)$$

$$\geq \Big(\frac{\tau V_{\mathcal{F}}}{\lambda}\Big)^2 \Big(\frac{1}{5}S'\Big)$$

$$\geq \Big(\frac{\tau V_{\mathcal{F}}}{\lambda}\Big)^2 \Big(\frac{1}{5} \times \frac{\lambda}{5}\Big)$$

$$= \Big(\frac{(\tau V_{\mathcal{F}})^2}{25\lambda}\Big).$$

So, a $2\delta$-separation implies

$$(2\delta)^2 = \frac{(\tau V_{\mathcal{F}})^2}{25\lambda} \implies \lambda = \Big(\frac{\tau V_{\mathcal{F}}}{10\delta}\Big)^2.$$

Then, we can 1. use the result of Lemma A.10 that $\log(\#\mathcal{F}_0)$ denotes as the log of the cardinality of $\mathcal{F}_0$ is at least $\log\binom{S}{\lambda} \geq (\lambda/4)\log(S)$ and 2. plugin the value of $S$ that gives

$$\log(\#\mathcal{F}_0) \geq \Big(\frac{\tau V_{\mathcal{F}}}{20\delta}\Big)^2 \log(S)$$

$$= \Big(\frac{\tau V_{\mathcal{F}}}{20\delta}\Big)^2 \log(d).$$

Based on the formula $\lambda = (\tau V_{\mathcal{F}}/10\delta)^2$, when $V_{\mathcal{F}}$ is fixed, it is evident that as $\delta$ decreases, $\lambda$ increases. Moreover, since $\lambda \leq d/10$, we need to assume that $d$ is large enough. Consequently, for small values of $\delta$, a sufficiently wide network becomes necessary. This observation is particularly interesting as it provides valuable insights into selecting an appropriate width for the network based on the input dimension. The larger the input dimension $d$, the wider the network should be.

STEP 2: DERIVING A LOWER BOUND FOR $\log(\#\mathcal{F}_{\mathcal{B}_{\mathrm{L}}})$

For the second step, our aim is to lower bound the log of the cardinality of the function class $\mathcal{F}_{\mathcal{B}_{\mathrm{L}}}$ using the result of the first step. Since we define $\mathcal{F}_0$ as a subclass of $\mathcal{F}_{\mathcal{B}_{\mathrm{L}}}$, we can conclude that the lower bound established for $\log(\#\mathcal{F}_0)$ in the first step also serves as a lower bound for $\log(\#\mathcal{F}_{\mathcal{B}_{\mathrm{L}}})$. We then can get

$$\log \mathcal{M}\big(2\delta, \mathcal{F}_{\mathcal{B}_{\mathrm{L}}}, \|\cdot\|_{L_2}\big) \geq \Big(\frac{\tau V_{\mathcal{F}}}{20\delta}\Big)^2 \log(d),$$

as desired.

**Remark B.1 (Constructing a vector in $\mathcal{A}$ using non-negative weights)** *Consider an example where $L = 3$, width of the network is set to four, and $\lambda = 2$. Our aim is constructing a binary vector with the length four (where $\lambda$ of those are ones) by a factor of $\tau \in (0, \infty)$. To get our desired final vector, the weights could be as follows:*

$$W^3 = [0 \quad \boldsymbol{c} \quad 0 \quad \boldsymbol{c}], \ W^2 = W^1 = \begin{bmatrix} 0 & 0 & 0 & 0 \\ 0 & \boldsymbol{c} & 0 & 0 \\ 0 & 0 & 0 & 0 \\ 0 & 0 & 0 & \boldsymbol{c} \end{bmatrix}.$$

*Then*

$$
W^3 W^2 W^1 = [0 \quad \boldsymbol{c} \quad 0 \quad \boldsymbol{c}]
\begin{bmatrix}
0 & 0 & 0 & 0 \\
0 & \boldsymbol{c} & 0 & 0 \\
0 & 0 & 0 & 0 \\
0 & 0 & 0 & \boldsymbol{c}
\end{bmatrix}
\begin{bmatrix}
0 & 0 & 0 & 0 \\
0 & \boldsymbol{c} & 0 & 0 \\
0 & 0 & 0 & 0 \\
0 & 0 & 0 & \boldsymbol{c}
\end{bmatrix}
= \tau \, [0 \quad \mathbf{1} \quad 0 \quad \mathbf{1}] \, .
$$

*Now, for example if we set $\boldsymbol{c} = 1/2$, then we get $\tau = 1/8$.*

∎

## B.2 PROOF OF LEMMA A.3

**Proof** To calculate the KL divergence between two normal distributions $\mathbb{P}_{f_{\Theta^j}}$ and $\mathbb{P}_{f_{\Theta^k}}$ of a continuous random variable, each with the corresponding densities $p_{f_{\Theta^j}}(\boldsymbol{z})$ and $p_{f_{\Theta^k}}(\boldsymbol{z})$, for all $j, k \in [\mathcal{M}]$ where $j \neq k$, we can 1. use the definition of the KL divergence, 2. plug the value of $p_{f_{\Theta^j}}(\boldsymbol{z})$ and $p_{f_{\Theta^k}}(\boldsymbol{z})$ in, 3. perform some simplification, 4. apply the definition of expected value, 5. the linearity of expected value, 6. use $y = f_{\Theta^j}(\boldsymbol{x}) + u$, 7. perform further rewriting, 8. apply the linearity of expected value, assuming independence between each $u_i$ and $\boldsymbol{x}_i$, 9. cancel out the second term ($\mathbb{E}[u] = 0$) and 10. recognize that only $\boldsymbol{x}$ values remain, to get

$$
D_{\mathrm{KL}}(\mathbb{P}_{f_{\Theta^j}} \parallel \mathbb{P}_{f_{\Theta^k}}) = \int_{\mathcal{X} \times \mathcal{Y}} p_{f_{\Theta^j}}(\boldsymbol{z}) \log \frac{p_{f_{\Theta^j}}(\boldsymbol{z})}{p_{f_{\Theta^k}}(\boldsymbol{z})} d\boldsymbol{z}
$$

$$
= \int_{\mathcal{X} \times \mathcal{Y}} p_{f_{\Theta^j}}(\boldsymbol{z}) \log \left( \frac{(1/\sqrt{2\pi\sigma^2}) e^{-\left((y - f_{\Theta^j}(\boldsymbol{x}))^2/2\sigma^2\right)} h(\boldsymbol{x})}{(1/\sqrt{2\pi\sigma^2}) e^{-\left((y - f_{\Theta^k}(\boldsymbol{x}))^2/2\sigma^2\right)} h(\boldsymbol{x})} \right) d\boldsymbol{z}
$$

$$
= \int_{\mathcal{X} \times \mathcal{Y}} p_{f_{\Theta^j}}(\boldsymbol{z}) \frac{1}{2\sigma^2} \left( \left(y - f_{\Theta^k}(\boldsymbol{x})\right)^2 - \left(y - f_{\Theta^j}(\boldsymbol{x})\right)^2 \right) d\boldsymbol{z}
$$

$$
= \mathbb{E}_{\boldsymbol{z} \sim p_{f_{\Theta^j}}(\boldsymbol{z})} \left[ \frac{1}{2\sigma^2} \left( \left(y - f_{\Theta^k}(\boldsymbol{x})\right)^2 - \left(y - f_{\Theta^j}(\boldsymbol{x})\right)^2 \right) \right]
$$

$$
= \frac{1}{2\sigma^2} \mathbb{E}_{\boldsymbol{z} \sim p_{f_{\Theta^j}}(\boldsymbol{z})} \left[ \left(y - f_{\Theta^k}(\boldsymbol{x})\right)^2 - \left(y - f_{\Theta^j}(\boldsymbol{x})\right)^2 \right]
$$

$$
= \frac{1}{2\sigma^2} \mathbb{E}_{\boldsymbol{z} \sim p_{f_{\Theta^j}}(\boldsymbol{z})} \left[ \left(f_{\Theta^j}(\boldsymbol{x}) + u - f_{\Theta^k}(\boldsymbol{x})\right)^2 - (u)^2 \right]
$$

$$
= \frac{1}{2\sigma^2} \mathbb{E}_{\boldsymbol{z} \sim p_{f_{\Theta^j}}(\boldsymbol{z})} \left[ \left(f_{\Theta^j}(\boldsymbol{x}) - f_{\Theta^k}(\boldsymbol{x})\right)^2 - 2u\left(f_{\Theta^j}(\boldsymbol{x}) - f_{\Theta^k}(\boldsymbol{x})\right) \right]
$$

$$
= \frac{1}{2\sigma^2} \left( \mathbb{E}_{\boldsymbol{z} \sim p_{f_{\Theta^j}}(\boldsymbol{z})} \left[ \left(f_{\Theta^j}(\boldsymbol{x}) - f_{\Theta^k}(\boldsymbol{x})\right)^2 \right] \right.
$$

$$
\left. - 2\mathbb{E}[u] \mathbb{E}_{\boldsymbol{z} \sim p_{f_{\Theta^j}}(\boldsymbol{z})} \left[ f_{\Theta^j}(\boldsymbol{x}) - f_{\Theta^k}(\boldsymbol{x}) \right] \right)
$$

$$
= \frac{1}{2\sigma^2} \left( \mathbb{E}_{\boldsymbol{z} \sim p_{f_{\Theta^j}}(\boldsymbol{z})} \left[ \left(f_{\Theta^j}(\boldsymbol{x}) - f_{\Theta^k}(\boldsymbol{x})\right)^2 \right] \right)
$$

$$
= \frac{1}{2\sigma^2} \int_{\boldsymbol{x} \in \mathcal{X}} \left(f_{\Theta^j}(\boldsymbol{x}) - f_{\Theta^k}(\boldsymbol{x})\right)^2 h(\boldsymbol{x}) d\boldsymbol{x} \, .
$$

Furthermore, by combining this result with Lemma A.2's result, it holds that for all $j, k \in [\mathcal{M}]$ and $j \neq k$

$$
D_{\mathrm{KL}}(\mathbb{P}^n_{f_{\Theta^j}} \parallel \mathbb{P}^n_{f_{\Theta^k}}) = \frac{n}{2\sigma^2} \int_{\boldsymbol{x} \in \mathcal{X}} \left(f_{\Theta^j}(\boldsymbol{x}) - f_{\Theta^k}(\boldsymbol{x})\right)^2 h(\boldsymbol{x}) d\boldsymbol{x} \, ,
$$

as desired.

Note that we only assume Gaussian noise, and accordingly we define $p_{f_{\Theta^j}}(\boldsymbol{z})$ as the joint density of $(\boldsymbol{x}, y)$ with regression function $f_{\Theta^j}$ (Equation 7). More specifically, the outputs of the network are not Gaussian: they are just Gaussian conditional on $\boldsymbol{x}$. ∎

### B.3 PROOF OF LEMMA A.4

**Proof** Consider a family of distributions $\{\mathbb{P}_{f_{\Theta^1}}, \ldots, \mathbb{P}_{f_{\Theta^{\mathcal{M}}}}\}$, then $I(J; Y|X)$ with respect to $J \to f_{\Theta^J} \to (Y|X)$, can be defined by using the KL divergence —as the underlying measure of distance— Wainwright (2019, Equation 15.29)

$$I(J; Y|X) := D_{\mathrm{KL}}\big(\mathbb{Q}_{(X,Y),J} \parallel \mathbb{Q}_{(X,Y)}\mathbb{Q}_J\big),$$

where, $\mathbb{Q}_{(X,Y),J}$ is the joint distribution of the pair $((X,Y), J)$, and $\mathbb{Q}_{(X,Y)}\mathbb{Q}_J$ is the product of their marginals, and assume that $\overline{\mathbb{Q}} \equiv \mathbb{Q}_{(X,Y)} := 1/\mathcal{M} \sum_{j=1}^{\mathcal{M}} \mathbb{P}_{f_{\Theta^j}}$ is the mixture distribution. We can rewrite the joint distribution $\mathbb{Q}_{(X,Y),J}$ as follows:

$$\mathbb{Q}_{(X,Y),J} = \mathbb{Q}\big((X,Y) = (x,y), J = j\big) = \mathbb{Q}\big((X,Y) = (x,y) \mid J = j\big)\mathbb{Q}(J = j).$$

Given that $J$ is chosen uniformly from $\{1, \ldots, \mathcal{M}\}$, we have:

$$\mathbb{Q}(J = j) = \frac{1}{\mathcal{M}}.$$

So, we can rewrite the joint distribution as follows:

$$\mathbb{Q}_{(X,Y),J}\big((x,y), j\big) = \mathbb{Q}\big((X,Y) = (x,y) \mid J = j\big)\mathbb{Q}(J = j)$$

$$= \mathbb{P}_{f_{\theta_j}}(x,y) \times \frac{1}{\mathcal{M}}.$$

We can 1. use the definition of KL divergence, 2. plug the value for the joint distribution into it and 3. invoke the definition of KL divergence and perform some rewriting to get

$$D_{\mathrm{KL}}\big(\mathbb{Q}_{(X,Y),J} \parallel \mathbb{Q}_{(X,Y)}\mathbb{Q}_J\big) = \sum_{(x,y),j} \mathbb{Q}_{(X,Y),J}\big((x,y), j\big) \log \frac{\mathbb{Q}_{(X,Y),J}\big((x,y),j\big)}{\mathbb{Q}_{(X,Y)}(x,y)\mathbb{Q}_J(j)}$$

$$= \sum_{(x,y),j} \frac{1}{\mathcal{M}}\mathbb{P}_{f_{\theta_j}}(x,y) \log \frac{\frac{1}{\mathcal{M}}\mathbb{P}_{f_{\theta_j}}(x,y)}{\mathbb{Q}_{(X,Y)}(x,y) \times \frac{1}{\mathcal{M}}}$$

$$= \frac{1}{\mathcal{M}} \sum_{j=1}^{\mathcal{M}} D_{\mathrm{KL}}\big(\mathbb{P}_{f_{\theta_j}} \parallel \mathbb{Q}_{(X,Y)}\big),$$

where, $\mathbb{Q}_{(X,Y)}$ is the mixture distribution. Accordingly, $I(J; Y|X)$ can be written in terms of component distributions $\{\mathbb{P}_{f_{\Theta^j}}, j \in [\mathcal{M}]\}$ as follows:

$$I(J; Y|X) = \frac{1}{\mathcal{M}} \sum_{j=1}^{\mathcal{M}} D_{\mathrm{KL}}\big(\mathbb{P}_{f_{\Theta^j}} \parallel \overline{\mathbb{Q}}\big).$$

Intuitively, it means the mean of the KL divergence between $\mathbb{P}_{f_{\Theta^j}}$ and $\overline{\mathbb{Q}}$— averaged over the choice of index $j$—gives the mutual information. Furthermore, based on the definition of the KL divergence, we can conclude that for $j = k$

$$D_{\mathrm{KL}}\big(\mathbb{P}_{f_{\Theta^j}} \parallel \mathbb{P}_{f_{\Theta^k}}\big) = 0.$$

Accordingly, we can 1. employ the mixture distribution formula in the above equation, 2. use the convexity of the KL divergence and apply Jensen inequality and 3. use the linearity property of sum to obtain

$$I(J; Y|X) = \frac{1}{\mathcal{M}} \sum_{j=1}^{\mathcal{M}} D_{\mathrm{KL}}\left(\mathbb{P}_{f_{\Theta^j}} \parallel \frac{1}{\mathcal{M}} \sum_{k=1}^{\mathcal{M}} \mathbb{P}_{f_{\Theta^k}}\right)$$

$$\leq \frac{1}{\mathcal{M}}\left(\sum_{j=1}^{\mathcal{M}}\left(\frac{1}{\mathcal{M}} \sum_{k=1}^{\mathcal{M}} D_{\mathrm{KL}}\big(\mathbb{P}_{f_{\Theta^j}} \parallel \mathbb{P}_{f_{\Theta^k}}\big)\right)\right)$$

$$= \frac{1}{\mathcal{M}^2} \sum_{\substack{j,k=1 \\ j \neq k}}^{\mathcal{M}} D_{\mathrm{KL}}\big(\mathbb{P}_{f_{\Theta^j}} \parallel \mathbb{P}_{f_{\Theta^k}}\big).$$

Consequently, if we can construct a $2\delta$-packing set such that all two distinct pairs of distributions $\mathbb{P}_{f_{\Theta^j}}$ and $\mathbb{P}_{f_{\Theta^k}}$ are close in average, then the mutual information can be controlled.

For the second claim, we employ the previous view with the result of Lemma A.2 to get

$$I(J; Y^n | X^n) \leq \frac{n}{\mathcal{M}^2} \sum_{\substack{j,k=1 \\ j \neq k}}^{\mathcal{M}} D_{\mathrm{KL}}\big(\mathbb{P}_{f_{\Theta^j}} \parallel \mathbb{P}_{f_{\Theta^k}}\big),$$

as desired. ∎

### B.4 PROOF OF LEMMA A.5

**Proof** The aim of this proof is to establish an upper bound on the mutual information $I(J; Y^n | X^n)$, for the $2\delta$-packing within the neural network's space $\mathcal{F}_{\mathcal{B}_{\mathrm{L}}}$. To achieve this, we invoke the result of Lemma A.4, which establishes a connection between the mutual information and the KL divergence. We then apply the result obtained from Lemma A.3. Finally, we employ the same re-scaling procedure as demonstrated in Wainwright (2019, Example 15.14) and Wainwright (2019, Example 15.16) to construct a $2\delta$-packing in such a way that, for a suitable constant $\kappa \in [1, \infty)$, we ensure that $\rho(f_{\Theta^j}(\boldsymbol{x}), f_{\Theta^k}(\boldsymbol{x})) \leq 2\kappa\delta$ holds for all pairs $f_{\Theta^j}(\boldsymbol{x})$ and $f_{\Theta^k}(\boldsymbol{x})$ corresponding to $j \neq k \in [\mathcal{M}]$.

We can 1. use the result provided by Lemma A.4, 2. use the fact that $\sum_{j,k=1}^{\mathcal{M}} D_{\mathrm{KL}}(\mathbb{P}_{f_{\Theta^j}} \parallel \mathbb{P}_{f_{\Theta^k}}) \leq \binom{\mathcal{M}}{2} \sup_{k,j}(D_{\mathrm{KL}}(\mathbb{P}_{f_{\Theta^j}} \parallel \mathbb{P}_{f_{\Theta^k}}))$ for all $j \neq k \in [\mathcal{M}]$, 3. calculate the permutation, 4. some arithmetic calculation, 5. use the fact that $\mathcal{M} \geq 1$, so $0 \leq (\mathcal{M} - 1)/\mathcal{M} < 1$, 6. use the view of Lemma A.3, 7. invoke the definition of $\rho$ as $L_2(\mathbb{P}_{\boldsymbol{x}})$- norm, 8. employ the re-scaling procedure and 9. simplify the factor 2 to obtain

$$
\begin{aligned}
I(J; Y^n | X^n) &\leq \frac{n}{\mathcal{M}^2} \sum_{\substack{j,k=1 \\ j \neq k}}^{\mathcal{M}} D_{\mathrm{KL}}\big(\mathbb{P}_{f_{\Theta^j}} \parallel \mathbb{P}_{f_{\Theta^k}}\big) \\
&\leq \frac{n}{\mathcal{M}^2} \binom{\mathcal{M}}{2} \sup_{\substack{j,k \in [\mathcal{M}] \\ j \neq k}} \Big(D_{\mathrm{KL}}\big(\mathbb{P}_{f_{\Theta^j}} \parallel \mathbb{P}_{f_{\Theta^k}}\big)\Big) \\
&= \frac{n}{\mathcal{M}^2} \frac{\mathcal{M}!}{(\mathcal{M}-2)!} \sup_{\substack{j,k \in [\mathcal{M}] \\ j \neq k}} \Big(D_{\mathrm{KL}}\big(\mathbb{P}_{f_{\Theta^j}} \parallel \mathbb{P}_{f_{\Theta^k}}\big)\Big) \\
&= \frac{n(\mathcal{M}-1)}{\mathcal{M}} \sup_{\substack{j,k \in [\mathcal{M}] \\ j \neq k}} \Big(D_{\mathrm{KL}}\big(\mathbb{P}_{f_{\Theta^j}} \parallel \mathbb{P}_{f_{\Theta^k}}\big)\Big) \\
&\leq n \sup_{\substack{j,k \in [\mathcal{M}] \\ j \neq k}} \Big(D_{\mathrm{KL}}\big(\mathbb{P}_{f_{\Theta^j}} \parallel \mathbb{P}_{f_{\Theta^k}}\big)\Big) \\
&= \frac{n}{2\sigma^2} \sup_{\substack{j,k \in [\mathcal{M}] \\ j \neq k}} \left(\int_{\boldsymbol{x} \in \mathcal{X}} \big(f_{\Theta^j}(\boldsymbol{x}) - f_{\Theta^k}(\boldsymbol{x})\big)^2 h(\boldsymbol{x}) d\boldsymbol{x}\right) \\
&= \frac{n}{2\sigma^2} \sup_{\substack{j,k \in [\mathcal{M}] \\ j \neq k}} \Big(\rho\big(f_{\Theta^j}(\boldsymbol{x}), f_{\Theta^k}(\boldsymbol{x})\big)^2\Big) \\
&\leq \frac{n(2\kappa\delta)^2}{2\sigma^2} \\
&= \frac{2n(\kappa\delta)^2}{\sigma^2},
\end{aligned}
$$

as desired. ∎

## B.5 PROOF OF LEMMA A.8

**Proof** In this proof, we first define a rotation matrix $\boldsymbol{R} \in \mathrm{SO}(d)$, which belongs to the special orthogonal group. Then, based on the fact that $\boldsymbol{R}^\top = \boldsymbol{R}^{-1}$, we can write $\boldsymbol{b}_k^\top \boldsymbol{x} = \boldsymbol{b}_k^\top \boldsymbol{R}^{-1} \boldsymbol{R} \boldsymbol{x} = (\boldsymbol{R}\boldsymbol{b}_k)^\top \boldsymbol{R}\boldsymbol{x}$. Accordingly, we can obtain

$$p(\boldsymbol{b}_k^\top \boldsymbol{x} > 0, \boldsymbol{0}, \boldsymbol{\Sigma}) = \frac{1}{(2\pi)^{d/2}\sqrt{|\boldsymbol{\Sigma}|}} \int_{(\boldsymbol{R}\boldsymbol{b}_k)^\top \boldsymbol{R}\boldsymbol{x} > 0} e^{\frac{-(\boldsymbol{R}\boldsymbol{x})^\top \boldsymbol{R}\boldsymbol{\Sigma}^{-1}\boldsymbol{R}^\top(\boldsymbol{R}\boldsymbol{x})}{2}} d\boldsymbol{x}\,.$$

By defining $\boldsymbol{G} := \boldsymbol{R}\boldsymbol{x}$ and $\tilde{\boldsymbol{b}}_k := \boldsymbol{R}\boldsymbol{b}_k$ and $d\boldsymbol{x} = (d\boldsymbol{x}/d\boldsymbol{G}) \times d\boldsymbol{G}$, we get

$$p(\boldsymbol{b}_k^\top \boldsymbol{x} > 0, \boldsymbol{0}, \boldsymbol{\Sigma}) = \frac{1}{(2\pi)^{d/2}\sqrt{|\boldsymbol{\Sigma}|}} \int_{\tilde{\boldsymbol{b}}_k^\top \boldsymbol{G} > 0} e^{\frac{-\boldsymbol{G}^\top \boldsymbol{R}\boldsymbol{\Sigma}^{-1}\boldsymbol{R}^\top \boldsymbol{G}}{2}} \left(\frac{d\boldsymbol{x}}{d\boldsymbol{G}}\right) \times d\boldsymbol{G}\,.$$

Then, 1. by setting $\tilde{\boldsymbol{\Sigma}} := \boldsymbol{R}\boldsymbol{\Sigma}^{-1}\boldsymbol{R}^\top$, $(d\boldsymbol{x}/d\boldsymbol{G}) := |\boldsymbol{R}|$ ($|\boldsymbol{R}| \equiv \det(\boldsymbol{R})$) and $\tilde{\boldsymbol{b}}_k = (\|\boldsymbol{b}_k\|_2, 0, \ldots, 0)^\top$, 2. by factoring out the term $|\boldsymbol{R}|$, 3. the fact that the probability density function of a Gaussian distribution for a random variable across its domain is 1 and 4. by noting that $|\boldsymbol{R}| = 1$, we can obtain

$$\begin{aligned} p(\boldsymbol{b}_k^\top \boldsymbol{x} > 0, \boldsymbol{0}, \boldsymbol{\Sigma}) &= \frac{1}{(2\pi)^{d/2}\sqrt{|\boldsymbol{\Sigma}|}} \int_{\tilde{\boldsymbol{b}}_k^\top \boldsymbol{G} > 0} e^{\frac{-\boldsymbol{G}^\top \tilde{\boldsymbol{\Sigma}}^{-1}\boldsymbol{G}}{2}} |\boldsymbol{R}| d\boldsymbol{G} \\ &= \frac{|\boldsymbol{R}|}{(2\pi)^{d/2}\sqrt{|\boldsymbol{\Sigma}|}} \int_{\tilde{\boldsymbol{b}}_k^\top \boldsymbol{G} > 0} e^{\frac{-\boldsymbol{G}^\top \tilde{\boldsymbol{\Sigma}}^{-1}\boldsymbol{G}}{2}} d\boldsymbol{G} \\ &= \frac{|\boldsymbol{R}|}{2} \\ &= \frac{1}{2}\,, \end{aligned}$$

as desired. ∎

## B.6 PROOF OF LEMMA A.9

**Proof** In this proof, our objective is to compute $\mathbb{E}[\phi(\boldsymbol{b}_k^\top \boldsymbol{x})\phi(\boldsymbol{b}_k^\top \boldsymbol{x})]$, which is a crucial component of the proof presented in Lemma 3.2, helping us establish a lower bound for a ReLU neural network. To achieve this, we can 1. employ the definition of expected value, 2. apply the definition of ReLU function, 3. perform some rewriting, 4. take out $\boldsymbol{b}_k^\top$ and $\boldsymbol{b}_k$, 5. employing Lemma A.8, 6. employ the definition of expectation, 7. apply the fact that $\mathbb{E}[\boldsymbol{x}\boldsymbol{x}^\top] = \boldsymbol{I}_d$, 8. apply $\boldsymbol{b}_k^\top \boldsymbol{I}_d \boldsymbol{b}_k = \boldsymbol{b}_k^\top \boldsymbol{b}_k$ and 9. use

$\boldsymbol{b}_k^\top \boldsymbol{b}_k = 1$ to obtain

$$
\begin{aligned}
\mathbb{E}\big[\boldsymbol{\phi}(\boldsymbol{b}_k^\top \boldsymbol{x})\boldsymbol{\phi}(\boldsymbol{b}_k^\top \boldsymbol{x})\big] &= \int_{\mathcal{X}} \boldsymbol{\phi}(\boldsymbol{b}_k^\top \boldsymbol{x})\boldsymbol{\phi}(\boldsymbol{b}_k^\top \boldsymbol{x})h(\boldsymbol{x})d\boldsymbol{x} \\
&= \int_{\boldsymbol{x}:\boldsymbol{b}_k^\top \boldsymbol{x}>0} (\boldsymbol{b}_k^\top \boldsymbol{x})^2 h(\boldsymbol{x})d\boldsymbol{x} \\
&= \int_{\boldsymbol{x}:\boldsymbol{b}_k^\top \boldsymbol{x}>0} (\boldsymbol{b}_k^\top \boldsymbol{x})(\boldsymbol{b}_k^\top \boldsymbol{x})^\top h(\boldsymbol{x})d\boldsymbol{x} \\
&= \boldsymbol{b}_k^\top \left( \int_{\boldsymbol{x}:\boldsymbol{b}_k^\top \boldsymbol{x}>0} \boldsymbol{x}\boldsymbol{x}^\top h(\boldsymbol{x})d\boldsymbol{x} \right) \boldsymbol{b}_k \\
&= \boldsymbol{b}_k^\top \left( \frac{\int_{\mathcal{X}} \boldsymbol{x}\boldsymbol{x}^\top h(\boldsymbol{x})d\boldsymbol{x}}{2} \right) \boldsymbol{b}_k \\
&= \boldsymbol{b}_k^\top \frac{\mathbb{E}[\boldsymbol{x}\boldsymbol{x}^\top]}{2} \boldsymbol{b}_k \\
&= \frac{1}{2} \boldsymbol{b}_k^\top \boldsymbol{I}_d \, \boldsymbol{b}_k \\
&= \frac{1}{2} \boldsymbol{b}_k^\top \boldsymbol{b}_k \\
&= \frac{1}{2},
\end{aligned}
$$

as desired. ∎

## C  EMPIRICAL DETAILS

Here, we first explain the two loss functions employed for training the networks in both classification and regression datasets and then we provide more details about the implementation settings.

### C.1  LOSS FUNCTIONS:

In the training process of ReLU networks, we utilize two distinct loss functions implemented in `TensorFlow`: "mean_squared_error" for regression and "categorical_crossentropy" for classification. The details of these loss functions are as follows:

**Cross-entropy loss**   For classification purpose, we use (categorical) `Cross-entropy` and define it as (Murphy, 2012; Ho & Wooky, 2019)

$$
\ell_{CE}\left(f_{\boldsymbol{\Theta}}\right) := -\frac{1}{n} \sum_{k=1}^{m} \sum_{i=1}^{n} \Big( (y_i)_k \log p\big(f_{\boldsymbol{\Theta}}(\boldsymbol{x}_i), k\big) \Big),
$$

where $(y_i)_k$ is the $k$-th element of the one-hot vector of the target label for the $i$-th data sample and

$$
p(f_{\boldsymbol{\Theta}}(\boldsymbol{x}), k) := \frac{e^{(f_{\boldsymbol{\Theta}}(\boldsymbol{x}))_k}}{\sum_{i=1}^{m} e^{(f_{\boldsymbol{\Theta}}(\boldsymbol{x}))_i}},
$$

where $(f_{\boldsymbol{\Theta}}(\boldsymbol{x}))_k$ is the $k$-th output of a network indexed by $\boldsymbol{\Theta}$.

**Mean-squared error**   For regression, we use `Mean-squared` and define it as (Botchkarev, 2018)

$$
\ell_{MS}\left(f_{\boldsymbol{\Theta}}\right) := \frac{1}{n} \sum_{i=1}^{n} \sum_{k=1}^{m} \frac{\Big( (y_i)_k - \big(f_{\boldsymbol{\Theta}}(\boldsymbol{x})\big)_k \Big)^2}{m},
$$

where $(y_i)_k$, is the $k$-th element of the target vector and $(f_{\boldsymbol{\Theta}}(\boldsymbol{x}))_k$ is the $k$-th output of a network indexed by $\boldsymbol{\Theta}$.

## C.2 Experimental setting:

Here we provide further details about our implementations. We itemized these properties such as accessing the dataset, the computer resources, the environment, splitting data and so on.

1. Computer resources: we conducted some of the experiments in Python using Google Colab and some of them using the basic plan of `deepnote` (https://deepnote.com). For the regression dataset, we used the basic plan of them that utilizes a machine with 5GB RAM and 2vCPU. For the `CIFAR10` dataset, we used one of the deepnote's plans that utilizes a machine with 16GB RAM and 4 vCPUs. It took about 8 hours. In fact, the computation time taken is not of great importance for this paper.

2. Test and train splitting: for classification, we have worked with well-known datasets like `MNIST`, `CIFAR10`, and `Fashion-MNIST`, utilizing the packages that import those datasets. Accordingly, our test and training data are exactly those provided by these datasets. For example, we imported the `Fashion-MNIST` dataset from `tensorflow.keras.datasets` package. For the housing dataset, we randomly divided the whole dataset, using $15\,000$ samples for training and the remaining samples for testing. Apart from this, as our aim in this paper is to determine which of these curves can better model the generalization error as the number of training data increases, we run the experiments (for example for $MNIST$ dataset) in intervals of 500 samples, that is, we first use 500 samples, then 1000 samples, then 1500 samples, and so forth. This choice of intervals gives us a detailed yet computationally feasible curve of the networks' performance as a function of the sample size. To ensure a significant amount of randomness multiple times, we randomly selected the required subset from our full training data. In our paper, the main trend of the test error as the number of training data increases is much more important than the specific error values themselves.

3. Optimization method: in the training procedure for our experiments, we have used `Adam` optimization method.

4. Finding the values for the two curves $(c_2 + \beta/n)$ and $(c_1 + \alpha/\sqrt{n})$: we use `SLSQP` method to find the parameters of those curves.

