# OpenReview forum: "How many samples are needed to train a deep neural network?"
_ICLR.cc/2025/Conference — ICLR 2025 Poster_

### Official Review · Reviewer_5XLe · 2024-11-04

**Soundness:** 4
**Presentation:** 4
**Contribution:** 3
**Rating:** 8
**Confidence:** 4

**Summary:**

The authors establish a lower bound on the packing number for deep ReLU networks with L1 constraints on the weights. Then they use the packing number bound and Fano’s inequality to provide a minimax risk lower bound that scale as sqrt(log(d)/n).

The rate of 1/sqrt(n) has been established in a few other settings for neural networks (shallow feedforward networks, linear networks), but not for deep neural networks with non-linear activations.

One of the main implications is the separation between this bound and the known minimax rates for linear methods which scale as 1/n.

 The authors provide some empirical support to their analysis, by experimenting with small datasets and networks, fitting O(1/n) and O(1/sqrt(n)) curves and showins that the latter achieves smaller residuals.

The question remains open on whether with specific data-dependent and algorithm-dependent assumptions 1/n is feasible in practical applications of neural networks.

**Strengths:**

Importantly the bound depends on the logarithm of the input dimension, d, and scales as 1/sqrt(n) matching comparable upper bounds. While this type of bound exists before, it had not been established for deep, non-linear networks.

Great clarity, organization and writing.

**Weaknesses:**

1. Some of the writing is missing clarity and could be rewritten in a more straightforward fashion.
- The sentences in lines 38-43
- The sentences in lines 120-123.

2. [Minor] In your proof sketch of Section 4, please properly introduce random variable J (it is only introduce in the appendix).

**Questions:**

1. Can you come up with some conditions under which the lower bound O(1/sqrt(n)) is actually not valid? For example, what is your intuition about the optimal rate for different architectures? What do you expect for the case when \sigma=0?

---

> ### Author Response · Authors · 2024-11-22
> **Official Comment by Authors**
>
> We are glad that you found this work interesting and greatly appreciate your instructive points and questions. We have addressed each of them as follows:
>
> $\textbf{Weaknesses:}$
>
> 1- Rewriting the lines 38-43: Thanks for raising this comment. We have rewritten these sentences to convey our goal more effectively and more straightforward as follows [lines 37-42]:
>
> A common limitation of the works mentioned above is that their generalization bounds often exhibit strong dependence&mdash;frequently exponential&mdash;on either the depth or width of the network. [Golowich, 2018] address this by providing bounds that avoid direct dependence on network depth for norm-constrained neural networks, with rates that grow only polynomially in depth. More recently, [Taheri, 2021; Mohades, 2023; Lederer, 2023] have proposed generalization bounds for regularized neural networks that show logarithmic growth in the total number of parameters, with the potential to decrease with additional layers, in contrast to exponential growth.
>
> 2- Rewriting the lines 120-123: Thanks for raising this comment. We have rewritten these sentences to convey our goal more effectively and more straightforward as follows [lines 120-124]:
>
> To the best of our knowledge, our work is the first to illustrate, both theoretically and practically, that in the absence of additional assumptions, the rate of $1/\\sqrt{n}$ is the "correct rate" in deep learning. It is important to note that some works, such as [Schmidt-Hieber, 2020], obtain a rate of $1/n$, but only under very strict assumptions. For instance, to achieve a rate of $1/n$ in their results, it is necessary to assume that the parameters are bounded by one and the network functions are bounded.
>
> We appreciate your careful reading.
>
> 3- Very good point. We will introduce the random index $J$ in Section 4 too (lines 297-298). Thank you for your precise insights and thorough review of the sections.
>
> $\textbf{Questions}:$ Very interesting questions!
>
> **A)**  As far as we know there are some works, such as [Schmidt-Heiber, 2020] (in lines [122-124], we have pointed out about this work) that under strick assumptions and network structure can achieve the rate $1/n$. However, there are two points to consider:
>
>  1. Their setting is not realistic and is far from practical applications (under restrictive assumptions).
>
>  2. Our simulation results support otherwise.
>
> **B)** Regarding different architectures: In fact, in our simulation, we go beyond the feed-forward setting to include CNNs and achieve the rate $1/{\\sqrt{n}}$.
>
> **C)** Regarding the noiseless case ($\sigma=0$): While our work does not explicitly address the noiseless case and requires further investigation in this area, we have identified some sources that discuss the noiseless Lasso in linear regression, such as [Van de Geer, 2018].
> In [Van de Geer, 2018], it is shown that, under certain conditions, the prediction error of the Lasso is dominated by the prediction error of its noiseless counterpart (for both fixed and random design matrices). Specifically, in their Theorem 14, they derive an exact expression for the prediction error of the noiseless Lasso (with a fixed design matrix) and demonstrate that its rate grows almost same as the noisy case. This result is established under two key conditions: the Beta-min condition and the compatibility condition.
> We believe that studying the noiseless case presents a fascinating direction for future research. Additionally, we would greatly appreciate learning about any other relevant papers on this topic that the reviewer is aware of and that we might have overlooked.
>
> Thank you again for raising these interesting questions and for considering our updates.
>
> 1) Van de Geer, S., On Tight Bounds for the Lasso, 2018. [ https://jmlr.csail.mit.edu/papers/volume19/17-025/17-025.pdf ]

---

### Official Review · Reviewer_R7Hh · 2024-11-04

**Soundness:** 2
**Presentation:** 2
**Contribution:** 2
**Rating:** 6
**Confidence:** 3

**Summary:**

The paper focuses on analyzing the sample complexity of learning deep ReLU feed-forward networks that have a bounded elementwise $\ell_1$ norm. The authors show that the minimax rate scales as $1/\sqrt{n}$, where $n$ is the sample size, and back up their theoretical claims with experimental results.

**Strengths:**

-	The problem of understanding the sample complexity of learning neural networks is an Important research topic.
-	The research presents a minimax lower bound for deep neural networks, which seems to be new.

**Weaknesses:**

-	It is not very clear to me why one should compare the results with $1/n$ rate. Based on my understanding, to achieve $1/n$ rate one often needs more structural assumptions on the target function, such as the ground-truth is spare in spare linear regression setting. Given that there is no such structural assumption in the paper, I’m not sure why one would expect such $1/n$ rate to happen.
-	The considered function space is elementwise $\ell_1$ norm bounded, which is not a very commonly considered setting.
-	The lower bound has an exponential dependency on $L$, which is not ideal though it is understandable that there might be technical difficulties to remove this.
-	It is not very clear to me the experiment setup is the right way to verify the correctness of the bound. These bounds are asymptotic bounds, meaning that the rate are only meaningful when $n$ is large enough.
First, in the current scale of $n$, the difference between $\sqrt{n}$ and $n$ are not very significant.
Second, the way of fitting the curve seems to consider all the data points, including those relatively small $n$. I believe such curve could not correctly represent the correct decaying rate.
Moreover, I believe these bounds in fact also depend on the norm of weight matrices.  It is possible the norm increases when $n$ increases, which could also affect the rate.
-	The experiment results many also affect by the optimization-related factors, for example converging to a local minima. These factors related to optimization are not considered in the theory, so it seems hard to draw conclusions from experiments.

**Questions:**

See weakness section above.

---

> ### Author Response · Authors · 2024-11-22
> **Official Comment by Authors**
>
> Thank you for your valuable comments and questions. We appreciate your recognition of the importance of our work and would like to
> address the questions and identified weaknesses as outlined below.
>
> $\textbf{Weaknesses}:$
>
> * Thank you for raising this question. As you correctly pointed out, some works achieve the rate of $1/n$ for neural networks, but under restrictive settings and assumptions [Schmidt-Hieber, 2020]. On the other hand, studies have established upper bounds for the error of neural networks under milder settings and assumptions at a rate of $O(1/\\sqrt{n})$ [Taheri et al., 2021]. This raises an important open question: can the rate $1/\sqrt{n}$ be improved in general without assuming restrictive conditions?
> While a corresponding result has been formally proved for $\ell_1$-regularized linear regression [Dalalyan et al., 2017, Proposition4 and Van de Geer and Lederer, 2013], no such proof exists for neural networks. In this work, we demonstrate that such an improvement is not possible in general, proving that the rate $1/\\sqrt{n}$ cannot be enhanced to $1/n$.
> Thank you again for pointing this out!
>
>
> * Thanks for raising this point! In addition to our explanations (lines [154-159]) about our reason for choosing the $\ell_1$-norm based on its inherent advantages over other norms like $\ell_0$, [Neyshabur et al. 2015] and then [Parhi and Nowak, 2023] state the equivalence of weight decay in deep learning to the well-known group lasso on the network weights. This highlights its closeness to practical applications.
>
>
> * Thank you for raising this question. We believe there might be a need for further clarification on this point, and we have addressed it as follows: Although we observe an exponential dependence on $L$, the key is that the value of $v_s$ (recall that $v_s$ is the sum of the $\ell_1$-norms of all the weights except those in the first layer) can be set as desired, as we don't impose any threshold on $v_s$. Once $v_s$ is chosen to be less than $L$, the term decreases. More specifically, while the problem would arise in a scenario where we only have $(v_s)^L$, this is not the case in our work, as the division by $L$ effectively resolves the issue.
>
>
> * Thank you for raising this question. We believe there might be a need for further clarification on this point, and we have addressed it as follows: You are absolutely right that, in practice, we plot the curves for all the data points, note that our curves are $(c_1 +\alpha/\sqrt{n})$ and $(c_2 +\beta/{n})$ (a factor of the curves $1/\sqrt{n}$ and $1/n$). In our theory, we don't figure out which aspects of our theory and conditions would be interpreted as asymptotic? As we don't have any lower bound for the number of the training samples in our setting.
>
> * This is an interesting point you have raised. As you know, theory and practice complement each other, with each supporting and reinforcing the findings of the other, rather than merely illustrating the other. Furthermore, the spectrum of aspects covered in practice, such as dealing with local optima, is often not addressed in theory, which typically focuses on global optima instead. Additionally, theory always involves assumptions and constraints, and those imposed by theory often do not fully address real-world problems. This is why, in our study, we focus on experiments with real-world data, as they offer practical insights that are closer to actual applications. Thank you for bringing up this point.
>
> Thank you in advance for considering our clarifications and updates.
>
>  1) High-dimensional Statistics, MIT OpenCourseWare, 2015 [ https://ocw.mit.edu/courses/18-s997-high-dimensional-statistics-spring-2015/resources/mit18_s997s15_chapter5/ ].
>
>  2)  Bühlmann, P. and  van de Geer, S., "Statistics for High-Dimensional Data Methods, Theory and Applications", 2011 [ https://link.springer.com/book/10.1007/978-3-642-20192-9 ].
>
> 3) Neyshabur, B., Tomioka, R., and  Srebro, N., "In Search of the Real Inductive Bias: On the Role of Implicit Regularization in Deep Learning", in Proc. ICLR, 2015 [ https://arxiv.org/abs/1412.6614 ].
>
> 4)  Parhi, R. and  Nowak,R., "Deep Learning Meets Sparse Regularization: A Signal Processing Perspective", IEEE Signal Process. Mag., vol. 40, no. 6, pp. 63–74, 2023 [ https://ieeexplore.ieee.org/stamp/stamp.jsp?tp=&arnumber=10243466 ].
>
> 5) Van de Geer, S., and Lederer, J., "The Bernstein–Orlicz Norm and Deviation Inequalities", Probability Theory and Related Fields, 157, 225–250, 2013 [ https://link.springer.com/article/10.1007/s00440-012-0455-y ].
>
> 6)  Dalalyan, A., Hebiri, M., and Lederer, J.  "On the prediction Performance of the Lasso". Bernoulli, 23(1), 552–581, 2017 [ https://arxiv.org/abs/1402.1700 ].

---

> ### Comment · Reviewer_R7Hh · 2024-11-25
>
> Thank you for addressing my concerns. I have carefully reviewed the authors' response and the comments from other reviewers. However, I would like to respectfully point out a few points where I find myself unable to fully agree with the authors' explanations:
>
> - Relation to $\ell_1$ norm: While the connection to weight decay promoting sparsity (as referenced by the authors) is true, sparsity is often considered in the neuron space, implying that only a few neurons should be active, rather than in the parameter space, which is the focus of the current paper.
>
> - Curve fitting in experiments:
>
>     - When comparing the rates $1/n$ and $1/\sqrt{n}$, I believe it would be more meaningful to focus on the case where $n$ is large, as this is where the distinction between the two rates becomes clearer.
>
>     - Also, I'm not sure about the rationale for introducing the constant terms $c_1$ and $c_2$ in the curve fitting, especially since such terms do not appear in the theoretical bounds.
>
>     - Another concern is the inclusion of data points with small $n$. When $n$ is small, the variance in the training loss due to randomness in training can be significant, which could potentially skew the fitted curve.

---

> > ### Author Response · Authors · 2024-11-28
> > **Authors' reply to Reviewer R7Hh**
> >
> > We sincerely appreciate your time and effort in reviewing our paper and for raising important points. We are also very grateful that you carefully read through the responses and comments. Furthermore, we are pleased to have addressed the questions you raised. In the following, we take this opportunity to address the remaining points.
> >
> > $\textbf{Relation to }$ $\ell_1$-norm: Thank you for raising this point. We are then happy to provide further clarification by discussing two different concepts of sparsity in neural networks:
> >
> > 1. Connection sparsity: This refers to the scenario where only a subset of the "connection" among neurons are active--there is a small number of non-zero connections between neurons, which induces sparsity at the level of connections or weights [Alvarez and Salzmann, 2016; Changpinyo et al., 2017; Feng and Simon, 2017; Kim et al., 2016; Lee et al., 2008;  Scardapane et al., 2017]. As pointed out in several works, a standard method to achieve connection-sparsity is the vanilla $\ell_1$-regularizer [ Hebiri et al., 2025]. This regularizer is the deep learning equivalent of the lasso regularizer in linear regression [Tibshirani, 1996] and has received considerable attention recently [ Kim et al., 2016].
> > More specifically, the $\ell_1$-regularizer acts on each connection individually, reducing the number of active connections among neurons.
> >
> > 2. Node sparsity: This refers to the scenario where all "incoming" weights of several nodes are simultaneously set to zero. As a result, these nodes become entirely inactive (see the third network in Figure. 1 of Hebiri et al., 2025). Node sparsity can be considered a structured form of connection sparsity, where the inactivity of connections clusters around specific nodes.
> > This regularizer is the deep learning equivalent of the group-lasso regularizer in linear regression [Bakin, 1999] and has received some attention recently [Alvarez and Salzmann, 2016; Feng and Simon, 2017; Scardapane et al., 2017].
> > Statistical theory of group-type norm is also studied in the literature [Neyshabur et.al, 2015; Lederer, 2024].
> >
> > To summarize, the network space considered in our setting&mdash;specifically, connection sparsity&mdash;also encompasses node sparsity, as node sparsity represents a structured form of connection sparsity.
> >
> >
> > $\textbf{Curve fitting:}$
> >
> > * The first and third point:
> > Thank you for your valuable inputs. In line with our theory, which is not asymptotic and does not make any assumptions about the sample size, we have also included small sample sizes in our numerical results. However, you raised a good point&mdash;we could also complement these plots with larger sample sizes. We have now completed this task, and the new plots have been added to our paper. The corresponding explanations can be found in Lines [408–414].
> >
> >      You are also correct regarding the variance. On average, however, its effects will largely cancel out and  will not be problematic.
> >
> > * Terms $c_1,c_2$:
> > Thank you for raising this question! We have indeed considered the constants $c_1$​ and $c_2$ in the context of approximation error, particularly because we are working with real datasets. This point is also addressed in our empirical section (Lines [395-396]).
> >
> > Please let us know if you have further questions or if anything remains unclear! Thanks in advance.
> >
> > 1. Alvarez, J., Salzmann, M., 2016. "Learning the number of neurons in deep networks". In: Proc. NIPS. pp. 2270–2278.
> >
> >
> > 2. Changpinyo, S., Sandler, M., Zhmoginov, A., 2017. "The power of sparsity in convolutional neural networks". arXiv:1702.06257.
> >
> > 3. Liu, B., Wang, M., Foroosh, H., Tappen, M., Pensky, M., 2015. "Sparse convolutional neural networks". In: IEEE Int. Conf. Comput. Vis. Pattern Recognit. pp. 806–814.
> >
> > 4. Scardapane, S., Comminiello, D., Hussain, A., Uncini, A., 2017. "Group sparse regularization for deep neural networks". Neurocomputing 241, 81–89.
> >
> > 5. Feng, J., Simon, N., 2017. "Sparse-input neural networks for high-dimensional nonparametric regression and classification". arXiv:1711.07592.
> >
> > 6. Tibshirani, R., 1996. "Regression shrinkage and selection via the lasso". J. R. Stat. Soc. Ser. B. Stat. Methodol. 58 (1), 267–288.
> >
> >
> > 7. Kim, J., Calhoun, V., Shim, E., Lee, J.-H., 2016. "Deep neural network with weight sparsity control and pre-training extracts hierarchical features and enhances classification performance: Evidence from whole-brain resting-state functional connectivity patterns of schizophrenia". Neuroimage 124, 127–146.
> >
> > 8. Bakin, S., 1999. "Adaptive regression and model selection in data mining problems (Ph.D. thesis)". The Australian National University.
> >
> > 9. Neyshabur, B. , Tomioka, R.,  and Srebro, N.," Norm-based capacity control in neural networks". In COLT, volume 40, pp. 1376–1401, 2015.
> >
> > 10. Hebiri, M., Lederer,  J., Taheri, M., "Layer sparsity in neural networks", Journal of Statistical Planning and Inference.

---

> > > ### Comment · Reviewer_R7Hh · 2024-12-03
> > >
> > > Thanks for the additional response. I have increased my score.

---

> > > > ### Author Response · Authors · 2024-12-03
> > > > **Official comment by authors**
> > > >
> > > > It is well-appreciated.

---

### Official Review · Reviewer_BYd5 · 2024-11-04

**Soundness:** 3
**Presentation:** 3
**Contribution:** 3
**Rating:** 8
**Confidence:** 3

**Summary:**

The paper establishes minimax lower bounds for deep ReLU feedforward neural networks in the context of regression. Of note, to establish such lower bounds, the paper establishes a lower bound on the packing number of a ReLU network function class.

**Strengths:**

The minimax lower bound for deep ReLU networks and the lower bound for the packing number of deep ReLU neural networks is novel and interesting.

**Weaknesses:**

*

**Questions:**

* Is it possible to get similar minimax rates if the input $x_i$ are subgaussian random vectors and/or noise are subgaussian?

* In view of lemma 5 of [1] (an upper bound on the generalization error) and theorem 2.2 in the paper, could a similar minimax lower bound be established for a function class consisting of more general architecture (e.g. other lipschitz activations, containing convolutions, etc) where the norm of the layers are bounded and the entire model is lipschitz?

[1] Mahsa Taheri, Fang Xie, & Johannes Lederer. (2020). Statistical Guarantees for Regularized Neural Networks.

---

> ### Author Response · Authors · 2024-11-22
> **Official Comment by Authors**
>
> We are glad that you found this work interesting and greatly appreciate your insightful questions. We have responded to each of them as follows:
>
> $\textbf{Questions}$:
>
> 1. Excellent point! This assumption is precisely what we need in certain parts of our proof, and further proof techniques must be employed to extend the results to other distributions, such as sub-Gaussian distributions. Intuitively, we think the results can also be extended to other distributions, though this would involve additional factors in the bounds depending on the specific properties of those distributions. Thanks for raising this thoughtful point.
>
> 2. Great question! We expect the same results, but the point is that some parts of our proofs involve using specific properties of ReLU functions, particularly the lemmas in Part 3 and the proof of Lemma 3.2. Therefore, we will investigate this in future work. Thanks for raising this question.

---

> > ### Comment · Reviewer_BYd5 · 2024-11-22
> >
> > Thank you for the response.

---

### Official Review · Reviewer_ABGY · 2024-11-08

**Soundness:** 3
**Presentation:** 3
**Contribution:** 2
**Rating:** 6
**Confidence:** 3

**Summary:**

This paper proposes to bound by below the minimax risk of a sub-class of ReLU Multi-Layer Perceptrons (MLP). This sub-class, denoted by $\mathcal{B}\_{\mathrm{L}}$, is built as the $\mathcal{L}^1$-ball of a given radius $v\_{\mathrm{s}}$ of the full space of parameters. The lower bound is obtained by using the packing number of $\mathcal{B}\_{\mathrm{L}}$.

**Strengths:**

# Originality

This paper attempts to find a lower bound for the minimax risk, which is not common in the literature.

# Clarity

The paper is easy to ready.

# Quality

The proof seems to be sound (I did not check everything).

**Weaknesses:**

# Originality

The paper uses almost the same tools as in [1], including the $\mathcal{L}^1$ regularization (which becomes the $\mathcal{L}^1$ ball in the space of parameters). The covering number is replaced with the packing number.

# Clarity

The motivation of finding such a lower bound is unclear: why would it be useful to get a lower bound on the minimax risk? What kind of information does it provide on the dataset, the function to approximate and the NN architecture?

Except it fits the framework of [1], what is the motivation for studying a subspace of MLPs defined by the $\mathcal{L}^1$ ball in the space of parameters? Why not choosing another metric?

# Significance

Unclear motivation (see Clarity).

# Quality

## Sharpness of the bound

According to the authors (see paragraph starting line 216), the lower bound is sharp, since it contains the same factors as the upper bound in [1]. Please note that the similarity between the two bounds is easily explained by the similarity of tools used.

But, very importantly, the bound in [1] is obtained with a loss with a strong $\mathcal{L}^1$ regularization term. Why would it be relevant to compare the two bounds?

## Interpretation of $V\_{\mathcal{F}}$

In paragraph starting line 216, the authors interpret $V\_{\mathcal{F}} = (v\_{\mathrm{s}} / L)^L$ as "a product over the $\mathcal{L}^1$-norm bounds of different layers in $\mathcal{F}\_{\mathcal{B}\_{\mathrm{L}}}$". This bound is obtained by performing an AM-GM inequality, which is known to be tighter when all the weight matrices have a similar $\mathcal{L}^1$ norm. However, it may be very loose otherwise, and the proposed interpretation tends to become inaccurate.

## Experiments

The authors have chosen to focus on experiments related to real-world data. This is undeniably useful, but the conditions of Theorem 2.2 are not fulfilled with such datasets. It would have been very useful to have results with a toy dataset, whose data are distributed in accordance to the assumptions (Gaussian inputs, outputs generated by a MLP, Gaussian noise...).


# References

[1] *Statistical guarantees for regularized neural networks*, Taheri et al., 2021.

**Questions:**

There is little discussion about the number $n$ of samples that are needed to train a NN (only at lines 233-238). What are typical values for $n$?

See above:
 * experiments on toy datasets matching the assumptions?
 * motivation for finding a lower bound?
 * relevance of the comparison with [Taheri et al., 2021]?

---

> ### Author Response · Authors · 2024-11-22
> **Official Comment by Authors (1)**
>
> Thank you for your valuable comments and suggestions on our paper. We appreciate the opportunity to address the points you raised, as outlined below (due to the character limit per official comment, we have provided our comments in two parts).
>
> $\textbf{Weaknesses:}$
>
>   *  $\textbf{Originality:}$
>   Thank you for raising this question. We believe there might be a need for further clarification on this point, and we have addressed it as follows: You are correct that our paper and the paper you mentioned [Taheri et al., 2021] indeed have the same "setup and setting"; however, there are significant differences in the main statistical goal and the theoretical tools employed:
>
>        **a)** The mentioned paper provides an upper bound, whereas in our work, we establish a mini-max lower bound.
>
>        **b)**  Although packing and covering are related concepts, we focus on "lower bounding" the packing number by totally different    techniques and tools opposed to upper bounding.
>
>        **c)** The proofs in our paper and theirs have no overlap; we use Fano's inequality and develop a lower bound for the packing of a  feed-forward neural network using entirely different methods, techniques and tools.
>
> *  $\textbf{Clarity:}$
> Thanks for pointing this out. In fact, in recent decades, finding mini-max lower bounds has become essential in learning theory and neural network research, with key works like [Raskutti and Wainwright, 2012] and [Wainwright, 2019] and etc. These bounds set a theoretical limit on the "best performance" any method can achieve on a given problem (see Chapter 5 of {High-dimensional Statistics}, MIT OpenCourseWare, 2015)
>
>    To clarify, a lower bound represents the smallest possible error that can be achieved, while a mini-max lower bound identifies this error for the most challenging case within a specific function class. This is particularly useful because, although upper bounds on error can often be tight (achievable by certain methods), lower bounds are typically harder to determine precisely. This often leaves a "gap," making a meaningful lower bound especially valuable for understanding the fundamental difficulty of a problem.
>
>     More significantly, mini-max lower bounds play a central role in neural networks and learning theory: they help determine the inherent difficulty of a learning problem, separating irreducible complexity from aspects that could be improved with better models. This optimality is precisely what we concluded when stating that the rate $1/\\sqrt{n}$ cannot be improved to $1/n$ in our conclusion section.
>
> *  $\textbf{Why}$ ${\\ell_1}$ $\textbf{-norm:}$
>     Thanks for raising this question! In addition to our explanations (lines [154-159]) about our reason for choosing the $\ell_1$-norm based on its inherent advantages over other norms like $\ell_0$, [Neyshabur et al. 2015] and then [Parhi and Nowak, 2023] state the equivalence of weight decay in deep learning to the well-known group lasso on the network weights.
>  This highlights its closeness to practical applications.
>
> *  $\textbf{Quality:}$
>     *  $\textbf{Sharpness of the bound:}$
>         Here, we would like to provide  two points:
>
>
>         **1.**  Although our problems and tools with [Taheri et al. 2021] are totally different, interestingly we reach almost the "same rates".
>
>         **2.**  In [Taheri et al. 2021], the $\ell_1$ regularization is not restrictive as they employ the concept of "scale regularization" to make sure they can cover the whole network space.
>             Thanks for raising these points.
>
> *  $\textbf{Interpretation of }$ $V_F:$
>        Thanks for raising this point! You are right about the inherent of AM-GM inequality. The key point in our work is that the value of $v_s$ (recall that $v_s$ is the sum of the $\ell_1$-norms of all the weights except those in the first layer) can be set as desired, as we don't impose any threshold on $v_s$. Once $v_s$ is chosen to be less than $L$, the term decreases. More specifically, while the problem would arise in a scenario where we only have $(v_s)^L$, this is not the case in our work, as the division by $L$ effectively resolves the issue.
>
> *  $\textbf{Experiments: }$ This is an interesting point you have raised. As you know, theory and practice complement each other, with each supporting and reinforcing the findings of the other, rather than merely illustrating the other (practice/theory). Additionally, theory always involves assumptions and constraints, and those imposed by theory often do not fully address real-world problems. This is why, in our study, we focus on experiments with real-world data, as they offer practical insights that are closer to actual applications. Thank you for raising this point.

---

> ### Author Response · Authors · 2024-11-22
> **Official Comment by Authors (2)**
>
> *  $\textbf{Questions: }$ Good point! We agree that, although this entire paper is implicitly related to the question of how many samples are needed to train a neural network, some explicit statements could provide further clarity. Therefore, we have added the following statements to address this point.
>
>       Based on Equation 6 (minimum number of samples), while we can't determine the exact required number of samples in advance, we can still make some useful observations.
>
>     **1.**  We need "many" samples based on the rate $1/\\sqrt{n}$.
>
>     **2.**  The noisier the data, the more training samples are needed to achieve the desired error level.
>
>      We have added these statements in lines [233-239]. Thanks for pointing this question out.
>
> *  $\textbf{Relevance of the comparison with [Taheri et al., 2021]: }$
>        Thanks for raising this question! You are right that a mini-max lower bound and an upper bound are fundamentally two different concepts. The mini-max lower bound identifies the worst-case performance limit that no method can surpass, while the upper bound specifies what a particular method or class of methods can achieve. In this work the purpose of comparing our work with [Taheri et al., 2021] is to assess the gap between these two bounds. This comparison reveals how closely the achievable performance (upper bound) approaches the lower bound. If the two bounds are close, it indicates that the problem is well-characterized, and there may be little room for improvement. On the other hand, a large gap suggests potential areas for further exploration, either by tightening the bounds or by improving the methods. Now, in this work as we find a mini-max lower bound with rate ($1/\sqrt{n}$), and compare it with the recent upper bounds (e.g. [Taheri et al., 2021]; thus anyone can readily conclude that the generalization error of a ReLU feed-forward neural network can't be improved beyond $1/\sqrt{n}$.
>
> Thank you in advance for considering our clarifications and updates.
>
> 1.  Tsybakov, A. B, "Introduction to Nonparametric Estimation". Springer Series in Statistics, 2004. [https://link.springer.com/book/10.1007/b13794 ]
>
> 2.  Raskutti, G.,  Wainwright, M.J., Yu, B.,  "Lower Bounds on Minimax Rates for Nonparametric Regression with Additive Sparsity and Smoothness", NIPS,2009. [ https://papers.nips.cc/paper_files/paper/2009/hash/b1563a78ec59337587f6ab6397699afc-Abstract.html ]
>
> 3.  Wainwright, M.J.,  "High-dimensional Statistics : A Non-asymptotic Viewpoint", Cambridge Uni , 2019.
> [ https://www.cambridge.org/core/books/abs/highdimensional-statistics/minimax-lower-bounds/B20391EAF07D3855BBFA5518AD61B99A ]
>
> 4.  High-dimensional Statistics, MIT OpenCourseWare, 2015. [ https://ocw.mit.edu/courses/18-s997-high-dimensional-statistics-spring-2015/resources/mit18_s997s15_chapter5/ ]
>
> 5.  Neyshabur, B., Tomioka, R., and  Srebro, N., "In Search of the Real Inductive Bias: On the Role of Implicit Regularization in Deep Learning", in Proc. ICLR, 2015. [ https://arxiv.org/abs/1412.6614 ]
>
> 6.  Parhi, R. and  Nowak,R., "Deep Learning Meets Sparse Regularization: A Signal Processing Perspective", IEEE Signal Process. Mag., vol. 40, no. 6, pp. 63–74, 2023. [https://arxiv.org/abs/2301.09554 ]

---

### Author Response · Authors · 2024-11-22
**General official comment**

The authors sincerely thank the reviewers for their time, effort, and thoughtful feedback on our paper. We hope our responses have effectively addressed their questions and concerns. To ensure clarity, we have provided detailed responses to each reviewer and incorporated the necessary revisions into the paper. For ease of reference, we have included corresponding line numbers in our comments to facilitate tracking the changes and ensure alignment with the reviewers' suggestions.

---

### Author Response · Authors · 2024-12-02
**General comment by authors**

Dear  reviewers,

Thank you very much for the interactions so far. If there are further questions about our rebuttal, we would be happy to address them.


Best regards,
Authors

---

### Meta-Review · Area_Chair_bbU4 · 2024-12-21

**Metareview:**

**(a) Summary**
This paper investigates the sample complexity required to train deep ReLU feed-forward neural networks. By establishing minimax lower bounds and leveraging Fano's inequality, the authors demonstrate that the generalization error scales at a rate of \(O(1/\sqrt{n})\) with the number of samples \(n\). The study also provides new insights into the role of packing numbers in quantifying neural network complexity and highlights a separation between these rates and the \(O(1/n)\) rates typically associated with linear models. Empirical experiments support the theoretical findings, showcasing how residuals align more closely with \(O(1/\sqrt{n})\) than \(O(1/n)\) in practical scenarios.

**(b) Strengths**
- Novel theoretical contributions, including a minimax lower bound for ReLU networks and new results on packing numbers.
- Clear and well-structured presentation, making the theoretical insights accessible.
- Empirical results that corroborate the theoretical claims, bridging the gap between abstract theory and practical observations.
- The paper addresses an important open question in learning theory, particularly in understanding the statistical limits of deep learning.

**(c) Weaknesses**
- The motivation for comparing with \(O(1/n)\) rates is not fully clarified; the structural assumptions needed for such rates are not directly addressed.
- The considered setting of elementwise \(L_1\)-bounded weights, while theoretically sound, is less common in practical neural network configurations.
- Some experimental choices (e.g., including small sample sizes and curve fitting) could be better justified or improved to reflect the theoretical asymptotics more robustly.
- The exponential dependency on the network depth in the bounds is acknowledged but not mitigated, which could be a concern for scalability.

**(d) Reasons for Acceptance**
The paper makes significant theoretical contributions to understanding the sample complexity of deep neural networks. Its findings on \(O(1/\sqrt{n})\) scaling are robust, novel, and well-supported by empirical evidence. The theoretical rigor combined with practical relevance justifies its acceptance despite minor weaknesses in experimental presentation and the scope of applicability of the \(L_1\)-bounded assumption. The work addresses a core issue in learning theory and contributes meaningfully to ongoing discussions about generalization in deep learning.

**Additional Comments On Reviewer Discussion:**

The discussion was productive, with authors addressing key concerns from reviewers:

1. **Reviewer ABGY** raised questions about the relevance of minimax lower bounds and the choice of \(L_1\) norms. The authors provided detailed justifications for the theoretical focus on lower bounds and clarified the practical motivations for \(L_1\)-norm constraints, referencing connections to group lasso regularization.

2. **Reviewer R7Hh** questioned the experimental setup, particularly the inclusion of small sample sizes and the choice of curve-fitting methods. The authors acknowledged the issues, updated the manuscript with larger sample size experiments, and provided explanations for the constants included in the fits.

3. **Reviewer BYd5** highlighted the potential to extend the results to more general architectures. The authors expressed openness to exploring this direction in future work and acknowledged the limitations of their current setting.

4. **Reviewer 5XLe** pointed out minor clarity issues in the writing and asked about conditions under which the lower bounds might fail. The authors revised the unclear sections and provided insightful responses about potential extensions and limitations.

The authors’ responses were comprehensive and improved the paper, addressing most concerns effectively. Based on this constructive dialogue and the reviewers’ eventual consensus on the paper's merit, I recommend its acceptance.

---

### Decision · Program_Chairs · 2025-01-22

Accept (Poster)